# Efficient Synthesis of Fluorescent Coumarins and Phosphorous-Containing Coumarin-Type Heterocycles via Palladium Catalyzed Cross-Coupling Reactions

**DOI:** 10.3390/molecules27217649

**Published:** 2022-11-07

**Authors:** Rumen Lyapchev, Ana I. Koleva, Iskra Z. Koleva, Kristian Subev, Ivelina Madzharova, Kristina B. Simeonova, Nevena Petkova-Yankova, Bernd Morgenstern, Vesela Lozanova, Petar Y. Petrov, Rositca D. Nikolova

**Affiliations:** 1Faculty of Chemistry and Pharmacy, Sofia University “St. Kliment Ohridski”, 1164 Sofia, Bulgaria; 2Department of Inorganic Solid-State Chemistry, Saarland University, 66123 Saarbrücken, Germany; 3Department of Medicinal Chemistry and Biochemistry, Medicinal University of Sofia, 1431 Sofia, Bulgaria

**Keywords:** 3-phosphonocoumarin, 1,2-benzoxaphosphorin, coumarin-3-carboxylates, Pd-catalyzed reactions, cross-coupling, Suzuki reaction, Sonogashira reaction, photophysical properties, fluorescence, DFT calculations

## Abstract

Quantum-chemical calculations on the spectral properties of some aryl substituted 3-phosphonocoumarins were performed, and the effect of the substituents in the aryl moiety was evaluated. The structures possessing promising fluorescent properties were successfully synthesized via Suzuki and Sonogashira cross-coupling. The synthetic protocol was also applied for the phosphorous chemoisomer of 3-phosphonocoumarin, 1,2-benzoxaphosphorin, and their carboxylate analogues. The optical properties of the arylated and alkynylated products were experimentally determined. The obtained quantum-chemical and experimental results give the possibility for a fine tuning of the optical properties of phosphorous-containing coumarin systems by altering the substituent at its C-6 position.

## 1. Introduction 

The presence of the coumarin structure in natural products and biologically active molecules has promoted considerable interest toward their synthesis [1,2,3]. The coumarin framework is widely used as a building block for obtaining derivatives that exhibit a wide variety of biological properties [4], especially anti-HIV, antitumor, anticoagulant, and antibiotic activities [5]. Moreover, some representatives of this class of compounds were screened as new drug candidates showing promising inhibition activity against the main protease of SARS-CoV-2 (PDB ID: 5N5O) [6,7].

Furthermore, the photophysical properties of coumarin-based fluorescent dyes have shown the advantage of using the benzopyran moiety as a fluorogenic scaffold [8]. Various coumarin-based compounds were applied in fields as laser dyes [9], cell-imaging biomarkers [10], and optical brighteners [11].

The synthesis of substituted coumarins is still dominated by classical methods, such as the Knoevenagel, Perkin, and Pechmann reactions [12,13,14], which although powerful and proven, have limited scope in terms of functional group compatibility. Recent studies have centered on the use of palladium-catalyzed cross-coupling C–C bond formation leading to the 3-, 4- and 6-substituted coumarins [15,16,17,18,19]; thus the number of Pd-catalyzed approaches for obtaining coumarins is constantly growing [20,21,22]. However, most of these synthetic protocols are focused on monosubstituted coumarins. Only limited applications of metal-catalyzed reactions for the synthesis of benzene ring-functionalized 3-substituted coumarins, especially when electron withdrawing groups are present, have been reported [23,24,25]. Therefore, herein we present an efficient method for the synthesis of fluorescent coumarins and phosphorous-containing coumarin-type heterocycles via palladium cross-coupling reactions, whereby the visible absorption properties of the compounds can be controlled by introducing various C-3 and C-6 substituents directly to the heteroaromatic structure.

## 2. Results and Discussion 

### 2.1. Theoretical Investigation of Substituted 3-Phosphonocoumarin Structures—Evaluation of the Fluorescent Properties 

As a part of our systematic investigation on the chemical behavior of 3-phosphonocoumarin, 1,2-benzoxaphosphorins as well as other 3-substituted coumarins, we theoretically studied the possibility to obtain substituted fluorescent species, where a “push–pull” effect with the substituent in the 3-rd position could be observed.

To check the photophysical properties and the possibility of fine tuning the spectral properties of phosphorous-containing model coumarins, quantum-chemical calculations of a series of phosphonocoumarins in acetonitrile as a solvent media were performed. For this purpose, density functional theory (DFT) [26,27,28,29] and time-dependent (TD) DFT [30,31] with the Gaussian16 suite of programs [32] were used.

The absorption and emission spectra of several compounds (**CM-1—CM-12**, Table 1), varying the substituent in position C-6 of the coumarin ring were calculated. Based on previous investigations in our group, coumarins bearing substituents in position C-6 [33] and C-7 [34] exhibit good fluorescent properties. The model compounds phenyl and aryl groups were chosen to enlarge the conjugated system, connected to the coumarin fragment. In order to take the solvent effect into account, an implicit solvent model (PCM) was used. The long-range corrected hybrid exchange-correlation functional CAM-B3LYP [35] paired with 6-31++G** basis set was employed [36].

The calculated absorption and emission energies of the phosphorous-containing coumarins were compared—structure **CM-1** and models having either EDG (electron-donating group) or EWG (electron withdrawing group) in 6-th position of the coumarin ring (**CM-2** to **CM-12**). The calculations showed that the chosen structures bearing different substituents red-shifted the absorption by 4–32 nm compared to the unsubstituted one (**CM-1**). This shift to a longer wavelength indicates that the differently substituted aryl groups have a significant impact on the absorbance of the coumarin derivatives.

The absorption spectra of structure **CM-5** with a p-methoxyphenyl group in position C-6 is more red-shifted compared to its *ortho*-isomer (**CM-3**), 320 nm versus 312 nm, respectively. For the model-bearing methoxy group in *meta* position, **CM-10**, the value for λ_abs_ is 309 nm. The absorption and the emission wavelengths of the model **C-10** are very similar to the non-substituted phenyl-bearing coumarin model **CM-2**. As expected, due to the lack of conjugation between the substituent and the coumarin moiety, an MeO-group in *meta* position would not affect the photochemical properties of the phosphorous-containing coumarin. This shows that not only the nature of the substituent in the phenyl ring is important, but also its position. It could be assumed that better conjugation occurs when there is an electron-donating group in *para*-position in the aryl substituent; therefore, the push–pull effect tends to be stronger.

In order to check whether a phenyl group (structure **CM-2**) could influence the spectral properties of the compound compared to the unsubstituted 3-phosphonocoumarin (**CM-1**), calculations of the absorption spectra data were performed. The λ_abs_ value for **CM-2** compound is 14 nm red-shifted compared to the one for **CM-1**; the same trend was observed for the emission of these molecules, Table 1. Therefore, a scrutinous selectivity of the substituents could efficiently be used to tune the fluorescent properties of the coumarins.

The effect of EWG in the aryl substituent was also considered. Two models having -CN or -F group were tested. The spectral properties of structure **CM-4**, containing CN-group in *ortho*-position, were investigated and the presence of this group lowers the λ_abs_ value, Table 1. When the CN-group is at *para* or *meta* position, **CM-11** and **CM-12**, the absorption maximum for both structures is slightly bathochromically shifted by 5 nm compared to the value for **CM-4** model, which is 299 nm. Comparing the calculated absorption of **CM-1** with **CM-6**, the 15 nm red-shift indicates that a minor change in the UV-spectra could be made when a fluorine atom is introduced in the aryl substituted.

Moreover, the effect of the alkyl groups in the aryl substituent (mesityl group), **CM-7**, was evaluated. The absorption wavelength of the compound is slightly red-shifted (only 4 nm) compared to **CM-1**. The presence of mesityl group (**CM-7**) would suppress the fluorescent properties compared to the phenyl substituent (**CM-2**), but still the emission is slightly red-shifted, compared to **CM-1**.

The possibility of having a longer conjugated system was also considered. Two compounds having triple C≡C bond were investigated (**CM-8** and **CM-9**). Comparing their emission to the one of **CM-1** (384 nm for **CM-8**, 402 nm for **CM-9,** and 344 nm for **CM-1**) indicates that the presence of an elongated π-system is beneficial to the fluorescent properties of the coumarin derivatives. In addition, (in **CM-9**) the presence of a methoxy group at *para*-position leads to higher λ_em_ compared to **CM-8**. Interestingly the emission for the model compounds **CM-2** and **CM-8** (having phenyl groups, 378 nm and 384 nm, respectively) and **CM-5** and **CM-9** (404 nm and 402 nm, respectively) slightly differ; therefore the addition of a triple bond does not affect the fluorescence properties of the coumarin species. The models bearing CN-group (**CM-4**, **CM-11,** and **CM-12**) exhibit emission maxima at a shorter wavelength (around 360 nm) compared to the structure having unsubstituted phenyl ring **CM-2**.

The calculated Stokes shifts for all the compounds were compared to the unsubstituted phosphorous coumarin **CM-1** (49 nm), Table 1. The absolute values of the calculated shifts for all the substituted compounds (**CM-2**–**CM-9**) are higher (64–84 nm) than the Stokes shift for **CM-1**. Compound **CM-4** bearing -CN group in the aryl moiety has the lowest calculated Stokes shift (58 nm) in comparison to all of the substituted coumarin derivatives. The highest one was calculated for structure **CM-5** that might be due to the EDG group in *para*-position in the aryl group.

The obtained theoretical results imply that the optical properties of phosphorous-containing coumarin systems can be tuned by altering the substituent in its C-6 position. These findings motivate us to synthesize the model structures that possess promising fluorescent properties and to experimentally check the spectral properties of the substituted 3-phosphonocoumarins.

### 2.2. Synthesis of Fluorescent Coumarins and Phosphorous Containing Coumarin-Type Heterocycles 

To obtain the calculated substituted fluorescent species, where a “push-pull” effect with the substituent in 3-rd position could be observed, a Pd-catalyzed cross-coupling reaction and in particulate the Suzuki–Miyaura coupling, was investigated.

The Knoevenagel reaction, as one of the classical cyclization methods, was the first tested approach to obtain the desired structures. For this purpose, the phenyl substituted salicylaldehyde **2** had to be synthesized. One of the possible reaction paths includes adding the desired phenyl fragment via the Suzuki coupling starting from 5-bromo-2-hydroxybenzaldehyde **1**, Figure 1. The reaction of compound **1** with phenylboronic acid was performed by using sodium carbonate decahydrate as a base in mixed solvent media (toluene:ethanol:water) and was catalyzed via bis(triphenylphosphine)palladium(II) dichloride (2 mol%) in inert atmosphere for 24 h at 80 °C. The yield of the arylated product **2** was moderate—69%. This compound was previously synthesized with slightly better yield using different base and solvent media, however, applying 10 mol% of the palladium catalyst [37].

The isolated 2-hydroxy-5-phenylbenzaldehyde **2** was used as a starting material in the Knoevenagel reaction with triethyl phosphonoacetate. It should be mentioned that compound **2** is not very stable—it slowly oxidized in air, both at room temperature or when stored at 4–10 °C. The condensation reaction was carried out following the procedure developed in our group [38] applying piperidine as a catalyst in dry toluene at reflux, Figure 2.

The reaction was completed in 5 h. Even though the reaction went smoothly, there were some difficulties related to the evaporation of the solvent and the separation of the two chemoisomers. The first obstacle was that compounds **3a** and **4a** have similar R_f_-values on both silica and alumina (TLC-monitoring); therefore, it was difficult to separate the compounds to sufficient purity by column chromatography. On the other hand, both compounds are slowly solidifying oils which made further purification by recrystallization impracticable.

It is interesting to note that the phenyl substituted benzopyrans **3a** and **4a** possess fluorescent properties as calculated. Therefore, a parallel approach for obtaining these structures was planned based on the preparation of 6-bromo-3-phosphonocoumarin **5** by Knoevenagel condensation of 5-bromo-2-hydroxybenzaldehyde **1** with triethyl phosphonoacetate [38], Figure 2, followed by a subsequent derivatization of the products by Suzuki coupling.

The advantages of this synthetic pathway are the usage of a stable and cheaper aldehyde and mainly the easier separation and further purification by column chromatography of products **5** and **6**.

Thereafter, the reaction of 6-bromo-3-phosphonocoumarin **5** with phenylboronic acid, Figure 3, was carried out using a relatively high load of palladium catalyst (3 mol%) due to the electron-donating oxygen atom from the lactone, bonded to the benzene ring in the coumarin moiety.

During the preliminary studies, some unexpected problems were faced due to the similar chromatographical behavior of compound **5** and the desired product **3a**. This complicated the TLC monitoring of the reaction and the purification of diethyl 6-phenylcoumarin-3-phosphonate **3a**. Formation of the dehalogenated coumarin structure **7** was also observed as a side product. Therefore, better reaction conditions needed to be developed for full conversion of **3b** where no side reaction could be observed.

The optimal conditions for the palladium-catalyzed coupling were determined through a series of experiments in which different combinations of catalyst (Figure 1) and solvent were applied. Two groups of palladium catalysts were employed, differing by the Pd oxidation state—Pd(0) and Pd(II), and by the coordinated ligand—phosphine and NHC-type.

Due to the drawbacks mentioned above, a series of small-scale experiments were performed, Figure 4, Table 2. The ratio of compounds **5**, **3a,** and **7** was monitored by NMR spectroscopy of the crude reaction mixture. Based on the chemical shift of the H-4 proton in all coumarin species **5**, **3a,** and **7,** the ratio of the compounds was determined.

For better comparison of the obtained results, the same reaction time was implied for all of the listed small-scale reactions (20 h). The reactions were performed using 1.2 equiv. of phenylboronic acid under an argon atmosphere at 80 °C. According to the literature data [39,40,41], addition of water as a co-solvent or other protic solvent to the mixture might improve the outcome of the reaction; thus, several combinations of water or ethanol with dry toluene or dioxane were also tested. Potassium carbonate (3 equiv.) was tested as a base for the reaction with the coumarin species due to the possibility of opening the lactone ring if stronger bases were used [42,43]. Not only did the replacement of the sodium carbonate with potassium carbonate improve the yields significantly but also K_2_CO_3_ is nontoxic, cheap, and strong enough to activate [44,45,46,47,48] all of the used Pd(II) catalysts. The results of the initial experiments are summarized in Table 2.

The catalysts were introduced as a solution in THF (0.5 mL) except for PdCl_2_(PPh_3_)_2_. Due to its insolubility, this catalyst was introduced to the reaction in its crystalline form. To properly evaluate the obtained results for PdCl_2_(PPh_3_)_2_, 0.5 mL THF was additionally added to the reaction.

As we mentioned above, the reaction was performed in the presence of different types of palladium catalyst according to the ligands coordinated to the metal atom. The catalytic activity of the phosphine type complexes PdCl_2_(PPh_3_)_2_ and Pd(PPh_3_)_4_ in the studied reaction differed significantly. The usage of the PdCl_2_(PPh_3_)_2_ in the arylation reaction led to almost full conversion of the starting 6-bromo-3-phosphonocoumarin **5**. As it could be seen from the listed results, Table 2, when employing bis(triphenylphosphine)palladium(II) dichloride, the reaction outcome is almost solvent-media-independent. However, not only the starting material but also the dehalogenated coumarin **7** was observed in the crude reaction mixture. The catalytic activity of tetrakis(triphenylphosphine)palladium(0) with the presented coumarin **5** was not satisfying—Pd(PPh_3_)_4_ complex gave the worst results amongst all of the chosen catalytic systems. However, when a mixed solvent (toluene:water or dioxane:water; entry 7 and entry 9, Table 2) was used, the yield of the desired product was increased.

The next set of experiments was performed by palladium complexes coordinating NHC-ligands. Two catalysts having bulkier ligands were tested in the Suzuki coupling—IMesPd(dmba)Cl and PEPPSI-type. The main observation in the IMesPd(dmba)Cl catalyzed reactions was the conversion and the yield of the targeted product. The selectivity of the reaction increased when water was introduced in the solvent media, entry 12 and entry 14, Table 2.

The full conversion of the 6-bromo-3-phosphocoumarin **5** into 6-phenyl-3-phosphonocoumarin **3a** was observed only when the reaction was carried out under the conditions listed as entry 17 and entry 19 (Table 2) in the presence of the PEPPSI-type catalyst.

It is interesting to note that when the reaction was performed in ethanol as a solvent media, a full conversion of the starting product was observed; however, the formation of **3a** was not detected. This statement was based on the NMR spectra where the characteristic signals for the protons at position C-4 for neither the **5**, **3a,** or **7** were found. This might be due to opening of the lactone ring in the presence of ethanol/base as it was previously reported [42,43,49,50,51,52].

Further scrutiny of the solvent media gave additional clarity of the overall conditions for better coupling reactivity. This made us consider the combination of PEPPSI-type catalyst and dioxane/water or toluene/water as the most effective in accomplishing high cross-coupling yield with full conversion of the starting material. The combination of toluene/water was preferred for the next Pd-catalyzed coupling reactions because of its low toxicity and price.

The structure of the 6-phenyl-3-phosphonocoumarin **3a** was determined by a single crystal X-ray analysis (CCDC 2209998, Figure 2) where only one substance was identified in the crystalline matter.

The cross-coupling reactivity of 6-bromo-3-phosphonocoumarin **5** was further tested under the above-optimized conditions, Figure 5. Aiming to increase the conjugation of the push–pull system, a series of boronic acids bearing electron donating and electron withdrawing groups were chosen, Table 3. The reactions were performed in five-time larger scales compared with the preliminary studies illustrated in Table 2.

Interestingly, under the optimized reaction conditions and in the presence of the selected boronic acid, the cross-coupling reaction with 6-bromo-3-phosphonocoumarin **5** could be achieved, Table 3, Figure 5. In cases of unsubstituted or *para*-substituted boronic acids, the yields of the desired products **3a**, **3c,** and **3d** varied from good to almost quantitative (61–95%) after purification by column chromatography. In all the presented experiments, the time needed for the full conversion of the coumarin **5** when substituted boronic reagents were used had increased compared with the unsubstituted phenylboronic acid. This implies that the hindrance of the reaction center and the electronic effect of the groups play a major role in the outcome of the reactions.

As a proof of that hypothesis, we observed that when the reaction center of the boronic reagent is hindered, the yield of the products **3b** and **3e** were low (entry 2 and entry 6, Table 3) using PEPPSI-type catalyst. The more hindered the boronic adduct is, the less yield is obtained. This observation is not new when a Suzuki reaction is performed with *ortho* or di-*ortho* substituted boronic species [53,54]; therefore, we decided to test other not so voluminous palladium catalysts that have shown good catalytic activity in the initial reaction with **5**. The arylation was performed with PdCl_2_(PPh_3_)_2_ (entry 3 and entry 7) thus resulting in significantly increasing of the yields for product **3b** and **3e**, from 41% to 63% and from 19% to 45%, respectfully.

Even when using less hindered catalyst, entry 7, Table 3, the reaction with mesityl boronic acid gave the lowest yields in comparison with the other arylated products. Thence, an experiment for obtaining **3e** with better yields was performed where the reaction temperature was increased to 100 °C; however, this did not give the positive outcome that we expected. The catalyst slowly decomposed, and only partial conversion of the starting material was indicated.

Phosphorus-containing coumarins such as **7 [1]** are of a great importance in the areas of *LiveScience* due to the similarity of phosphorous compounds to the naturally occurring carboxylic acid derivatives and their potential application in varies biological systems. In order to further explore the generality of our modified procedure and to obtain analogous structures, the chemical behavior of halogenated ethyl and methyl ester of coumarin-3-carboxylic acid **8a,b** was studied in the Suzuki reaction, Figure 6.

The same reaction conditions—PEPPSI-type catalyst, mixed solvent (toluene:water) and potassium carbonate as a base—were employed in for the reactions with ethyl and methyl ester of coumarin-3-carboxylic acid **8a,b**. A striking observation was the extreme reactivity of the bromo carboxylates in the Suzuki reaction catalyzed by the PEPPSI complex. For example, the time needed for full conversion of both starting materials, in case of unsubstituted or substituted in *para*-position boronic acid, was determined to be in the range of 30 to 60 min.

Longer reaction time and lower yields were observed again when 2-metoxyphenyl boronic acid was used as an adduct in the Suzuki reaction with coumarin **8a**, entry 3**,**
Table 4. The replacement of the bulky PEPPSI-type catalyst with less hindered complex as PdCl_2_(PPh_3_)_2_ resulted in increasing the yield from 20% to 68% (entry 4, Table 4). The usage of bis(triphenylphosphine)palladium(II) dichloride gave us the opportunity to obtain coumarin derivatives with even more hindered boronic acids—as 2,4,6-trimethylphenyl boronic acid and 2,6-dimethoxyphenyl boronic acid (entry 10–13, Table 4).

A weak reactivity of the coumarin species in the studied conditions in cases listed as entry 10–13 was observed. The obtained low yields for products **9e,f** and **10e,f** might be due not only to the fact that there was not full conversion of the esters but also to difficulties in their purification.

Longer reaction time and lower yields were observed again when 2-metoxyphenyl boronic acid was used as an adduct in the Suzuki reaction with coumarin **8a**, entry 3, Table 4. The replacement of the bulky PEPPSI-type catalyst with less hindered complex as PdCl_2_(PPh_3_)_2_ resulted in increasing the yield from 20% to 68% (entry 4, Table 4). The usage of bis(triphenylphosphine)palladium(II) dichloride gave us the opportunity to obtain coumarin derivatives with even more hindered boronic acids—as 2,4,6-trimethylphenyl boronic acid and 2,6-dimethoxyphenyl boronic acid (entry 10–13, Table 4).

The synthesis of compounds **9a,c,d** via the Suzuki reaction was previously reported by two different research collectives [23,24]; however, different catalytic systems were used. The obtained yields for these compounds were lower under longer reaction times. For example, Gobec et al. [24] obtained the ethyl 6-phenylcoumarin-3-carboxylate **9a** with only 33% yield, while the reaction time was not specified (overnight); Carbonaro et al. [23] reported the formation of product **9a** in 55% yield for 12–16 h. In the procedure that we developed, the ethyl 6-phenylcoumarin-3-carboxylate **9a** was isolated with 90% yield after only 30 min, entry 1 (Table 4). The yield of compound **9c** was reported to be 67% (overnight) [24] and 66% (12–16 h) [23]. Using the optimized procedure based on the PEPPSI-type catalyst, we managed to isolate the ethyl 6-(4-methoxyphenyl)coumarin-3-carboxylate **9c** in 92% for 45 min, entry 6 (Table 4). The yield of **9d** obtained from a Suzuki reaction catalyzed by Pd(PPh_3_)_4_ was reported [24] to be 54% (overnight). As can be seen from Table 4, changing the catalytic system could significantly increase the reactivity of the coumarin **8a**, thus resulting in the 80% yield for coumarin **9d** and the conversion time of 45 min, entry 8.

The reactivity of 6-bromocoumarin-3-carboxylates **8a,b** was tested with even bulkier adduct—2,6-dimetoxyboronic acid, entry 12 and entry 13, Table 4. Unfortunately, in both cases, no full conversion of the starting compounds was observed. Even though the reaction times were long, a complexed mixture was obtained, and none of the structures were isolated due to the absence of predominant product. Another drawback was the similar chromatographic properties of some components of the mixture on both silica and alumina.

Motivated by these results, we next sought to expand the scope of the optimized procedure to the arylation of a more challenging substrate, such as 1,2-benzoxaphosphorin **6**, Figure 7.

Under the optimized reaction conditions—PEPPSI-type catalyst, solvent mixture of toluene:water, potassium carbonate and in the presence of the selected boronic acid—the cross-coupling reaction with ethyl 6-bromo-2-ethoxybenzo[e][1,2]oxaphosphorine-3-carboxylate **6** was achieved, Figure 7, Table 5. The yields for the arylated products **11a–d** are similar to the ones obtained for the corresponding chemoisomeric structures **3a–d**. However, the time needed for the consumption of the starting material compared with the 3-phosphonocoumarin derivatives was shorter.

A different catalyst was used, Pd(PPh_3_)_2_Cl_2_ (entry 3, Table 5), only when 2-methoxyphenylboronic acid was employed. The bis(triphenylphosphine)palladium(II) dichloride complex gave better results with the studied carboxylates and with 6-bromo-3-phosphonocoumarin **5**. However, in the case of oxaphosphorine-3-carboxylate **6**, the yield of product **11b** was moderate (45%).

In the studied reaction conditions, poor reactivity of the coumarin species was observed with respect to hindered boronic acids; therefore, no experiments were performed with 2,4,6-trimethylphenylboronic acid and 2,6-dimethoxyphenylboronic acid. On the other hand, oxaphosphorine-3-carboxylate **6** is a byproduct, and there is still no efficient method for its preparation on a large scale.

The Suzuki reaction of ethyl 6-bromo-2-ethoxybenzo[e][1,2]oxaphosphorine-3-carboxylate **6** is applicable for boronic acids bearing electron-donating and electron-withdrawing groups. The only limitation of the procedure is when a hindered organoboron substrate is used.

To extend the π-system of the heterocyclic system, another Pd-catalyzed reaction was tested on the studied coumarins—the Sonogashira alkynylation, Figure 8, Table 6.

Most reactions with the chosen substrates were performed under classical conditions for Sonogashira coupling [55,56]—Pd(PPh_3_)_2_Cl_2_/CuI as a catalytic system in the presence of Et_3_N in DMF at 80 °C under argon atmosphere. To compensate the electron-rich nature of the 4-ethynylanisole, entry 4, Table 6, PEPPSI-type complex was applied. However, these conditions resulted in the formation of product **12d** in only a 56% yield. Reaction of the same alkynylating reagent and ethyl coumarin-3-carboxylate **8a** was not carried out since the product is a known compound and was prepared under the same classical conditions [57]. The Sonogashira reaction in analogues conditions involving the methyl carboxylate **8b** had also been reported [58].

Surprisingly, the reactions of ethyl 6-bromo-2-ethoxybenzo[e][1,2]oxaphosphorine-3-carboxylate **6** under the described conditions led to the formation of complexed reaction mixtures, and none of the structures **12c,e** were isolated due to the absence of predominant product.

### 2.3. Photophysical Properties 

The photophysical properties of the synthesized compounds were experimentally determined by UV-VIS and fluorescent spectroscopy. The absorption and fluorescence spectra of the aryl substituted coumarin, **3a,** were recorded in five different solvents—acetonitrile (MeCN), methanol (MeOH), ethyl acetate (EtOAc), dimethyl sulfoxide (DMSO), and dichloromethane (DCM), Figure 3, Table 7. The obtained results showed that the solvent slightly affects the position of the longest absorption wavelength band. For example, in the polar protic solvent (MeOH) the absorbance maximum was exhibited at 346 nm, and in the non-polar one—dichloromethane was hypsochromically shifted only by 2 nm (Table 7 and Figure 3). In the emission spectra obtained, MeOH and CH_2_Cl_2_ bands at 459 nm and at 452 nm, respectively, were observed. Interestingly, although DMSO is an aprotic solvent, the maximum in the fluorescence spectrum was bathochromically shifted only by 1 nm compared to MeOH. The biggest difference was noted in the case of EtOAc—the maximum was blue-shifted by 16 nm with respect to the corresponding in DMSO, which was the most red-shifted one.

The spectral data of the other three coumarins bearing phenyl substituents—**9a**, **10a** and **11a** dissolved in acetonitrile, methanol and dichloromethane demonstrated minor solvatochromic effect from 13 to 16 nm—in MeOH the bands were red-shifted, while in DCM they were blue-shifted, and the maximum in acetonitrile was always between these two maxima (Table 7, Figure 3). Due to the higher solubility of the coumarins in acetonitrile compared to methanol, MeCN was selected as a solvent for all other synthesized derivatives.

The UV-VIS and emission spectra of **3b–e**, **12a,** and **12d** compounds in acetonitrile are shown in Figure 4. As predicted by the DFT calculations, the presence of phenyl substituent in 6-th position of the coumarin ring (**3a**) leads to a redshift both in the absorption and the fluorescence spectra by 19 and 42 nm, respectively, compared to the unsubstituted substance. The absorption maximum in the UV spectrum of the **3e** bearing mesityl group, was at shorter wavelength like in the simulated one—333 nm in comparison with the corresponding value for **3a**—343 nm. The compounds having methoxy group in *ortho*-position and F-atom in the aryl substituent exhibited maximum at 345 nm, while for Ph-C≡C λ_max_ was slightly red-shifted by 6 nm. The compound containing methoxyphenyl group in *ortho*-position in the aryl substituent emitted at 493 nm.

The derivative **12a** with triple bond and without OMe group in *para* position exhibits a maximum at a shorter wavelength 461 nm. The largest bathochromic shift in the absorption and emission spectra was observed for the coumarin derivates with longer conjugated π-system, compounds **3c** and **12d**. They exhibited the same absorption maximum at 356 nm, which by 13 nm red-shifted compared to the corresponding spectra of the phenyl derivate, **3a**, while their emission is shifted to the longest wavelength by 82 and 90 nm for **3c** and **12d**, respectively.

The same trend was observed for the ethyl and methyl coumarin-3-carboxylates and benzoxaphosphorines—the absorption and emission bands’ shift was highest when there is *para*-methoxyphenyl substituent (Figure 5) as the fluorescence is very similar to that of the corresponding phosphonocoumarins **3c** and **12d**—546, 543, and 541 nm for **9c, 10c,** and **11c**, respectively, Figure 6. Thus, the substituent in position C-3 does not affect the fluorescence of the coumarin species. The λ_max_ in the fluorescence spectra of phenyl derivates is red-shifted by nearly 50 nm with respect to the unsubstituted coumarins, which for each series are denoted with the number of the parent series followed by H (Table 7).

The observed Stokes shift for all compounds with C-6 substituents was above 100 nm, while the highest was determined for the *para*-methoxyphenyl derivatives, and it varied from 179 for **3c** to 191 for **11c**. For the unsubstituted coumarins, the Stokes shift is smaller and is around 85 nm.

The computed maxima of the absorption spectra, multiplied by the scaling factor, is in very good agreement with the experimental data (Table 8). The differences do not exceed 5 nm. The order of λ_max_ in the calculated UV spectra is the same as in the experimentally obtained—in direction H-, mesityl-, Ph-, *ortho*-MeOC_6_H_4_-, F-, Ph-C≡C-, *para*-MeOC_6_H_4_-, and *para*-MeOC_6_H_4_-C≡C substituents at position C-6 is the observed bathochromic shift.

The variance for the fluorescence bands in some cases is much more pronounced by up to 62 nm for coumarin **12d**. However, in the results for **3a**, **3e,** and **12a,** the simulated data are red-shifted by only 1, 7, and 2 nm, respectively. The calculated Stokes shift differs by around 40–45 nm. As for the two structures with methoxy group in *para* position, **3c** and **12d,** the distinction is higher and is 95 and 112 nm, respectively.

To test the fluorescent properties of coumarins **3a** and **3c** in different pH, a series of experiments by varying the concentration of sulfuric acid and sodium hydroxide (Figure 7) were performed. Interestingly, the fluorescence of the *para*-methoxyphenyl derivative **3c** was suppressed when lowering the pH of the solution by adding sulfuric acid, while the emission of the phenyl derivative **3a** had increased. This different behavior of the coumarin species might be due to different protonation positions. In compound **3a,** the protonation could occur at the lactone oxygen atom therefore the conjugation would be more effective. The quenching of the fluorescence of coumarin **3c** might be due to the protonation of the MeO-group which decreases the donor ability of the oxygen atom. The addition of sodium hydroxide might lead to lactone ring opening, thus loss of conjugation (Figure 7c,d).

## 3. Materials and Methods 

Melting points were determined on an SRS MPA120 EZ-Melt apparatus and are used without correction. The IR spectra were recorded with a Shimadzu FTIR-8400S spectrophotometer. ^1^H, ^13^C, and ^31^P NMR spectra were recorded on a Bruker AVNEO 400 spectrometer (at 400 MHz for ^1^H, 100.6 MHz for ^13^C, 376.46 MHz for ^19^F, and 161.98 MHz for ^31^P, respectively), Bruker Avance III 500 spectrometer (at 500 MHz for ^1^H, 125.7 MHz for ^13^C and 202.4 MHz for ^31^P, respectively). Chemical shifts are given in ppm from tetramethylsilane as internal standard with CDCl_3_ as solvent. Chemical shifts in ^19^F NMR referenced by IUPAC recommendations (2000) and in ^31^P NMR spectra were referred from external standard—85% H_3_PO_4_. The NMR-spectra of the newly synthesized compounds could be found in the Appendix A. Liquid chromatography mass spectrometry analysis (LC-HRAM) was carried out on Q Exactive^®^ hybrid quadrupole-Orbitrap^®^ mass spectrometer (ThermoScientific Co, Waltham, MA, USA) equipped with a HESI^®^ (heated electrospray ionization) module, TurboFlow^®^ Ultra High Performance Liquid Chromatography (UHPLC) system (ThermoScientific Co, Waltham, MA, USA) and HTC PAL^®^ autosampler (CTC Analytics, Zwingen, Switzerland). The chromatographic separations of the analyzed compounds were achieved on Nucleoshell C18 (100 × 2.1 mm, 2.7 µm) analytical column (Macherey-Nagel, Düren, Germany) using gradient elution at 300 µL/min flow rate. The used eluent systems were: A—0.1% formic acid in water; B—0.1% formic acid in CH_3_CN. Full-scan mass spectra over the *m*/*z* range 100–600 were acquired in positive ion mode at resolution settings of 140,000. The used mass spectrometer operating parameters were: spray voltage—4.0 kV; capillary temperature—320 °C; probe heater temperature—300 °C; sheath gas flow rate 40 units; auxiliary gas flow 12 units; sweep gas 2 units (units refer to arbitrary values set by the Q Exactive Tune software); and S-Lens RF level of 50.00. Nitrogen was used for sample nebulization and collision gas in the HCD cell. All derivatives were quantified using 5 ppm mass tolerance filters to their theoretical calculated m/z values. Data acquisition and processing were carried out with XCalibur^®^ ver 2.4 software package (ThermoScientific Co, Waltham, MA, USA). UV-Vis spectra were carried out on a Shimadzu UV-1800. Fluoresce spectra were recorded at room temperature on a PerkinElmer LS45. Reactions were monitored by TLC on silica gel 60 F_254_. Column chromatography was carried out on silica gel (Merck 0.043–0.063 mm) using as eluent *n*-hexane/EtOAc mixture with increasing polarity. The X-ray data set was collected using a Bruker D8 Venture diffractometer with a microfocus sealed tube and a Photon II detector. Monochromated Mo_Kα_ radiation (λ = 0.71073 Å) was used. Data were collected at 133(2) K and corrected for absorption effects using the multi-scan method. The structure was solved by direct methods using SHELXT [59] and was refined by full matrix least squares calculations on F^2^ (SHELXL2018 [60]) in the graphical user interface Shelxle [61]. Refinement—all non H-atoms were located in the electron density maps and refined anisotropically. C-bound H atoms were placed in positions of optimized geometry and treated as riding atoms. Their isotropic displacement parameters were coupled to the corresponding carrier atoms by a factor of 1.2 (CH, CH_2_) or 1.5 (CH_3_). Disorder: The ethoxy-group on O4 was split over two positions. Its occupancy factors refined to 0.775 for the major compound.

Computational details—the quantum-chemical calculations of the series of phosphonocoumarins in acetonitrile were performed using density functional theory (DFT) [26,27,28,29], time-dependent (TD) DFT [30,31] with Gaussian16 suite of programs [32] and the long range corrected hybrid exchange-correlation functional CAM-B3LYP [35], paired with 6-31++G** basis set was employed and a polarizable continuum model (PCM) was used [36].

All chemical reagents were purchased from Merck and Sigma Aldrich. The starting 6-bromo-3-substitueted-2-oxo-2*H*-1-benzopyrans **5**,**6**,**8a,b** were prepared according to reported procedures [38]. The used Pd-complexes (PdCl_2_(PPh_3_)_2_ [62], IMesPd(dmba)Cl [45], and PEPPSI-type catalyst [44]) were prepared according to known procedures.

### 3.1. Starting Materials 


**
*4-Hydroxy-[1,1’-biphenyl]-3-carbaldehyde, 2 *
**


Compound **2** was prepared according to the reported procedure [37].

^1^H NMR (500 MHz, CDCl_3_) δ = 10.926 (s, 1H, OH), 9.889 (s, 1H, CHO), 7.672–7.781 (m, 2H, aromatic), 7.171–7.573 (m, 5H, aromatic), 7.000 (d, J = 8.4 Hz, 1H, aromatic).


**
*Diethyl (6-bromo-2-oxo-2H-chromen-3-yl)phosphonate, 5*
**


Compound **5** was prepared according to the reported procedure [38].

^1^H NMR (500 MHz, CDCl_3_) δ = 8.425 (d, J = 17.1 Hz, 1H, H-4), 7.714–7.728 (m, 2H, H-7, H-5), 7.26 (d, J = 9.4 Hz, 1H, H-8), 4.215–4.347 (m, 4H, two CH_3_CH_2_O), 1.387 (t, J = 7.1 Hz, 6H, two CH_3_CH_2_O).


**
*Ethyl 6-bromo-2-ethoxybenzo[e][1,2]oxaphosphinine-3-carboxylate 2-oxide, 6*
**


Compound **6** was prepared according to the reported procedure [38].

^1^H NMR (500 MHz, CDCl_3_) δ = 8.132 (d, J = 36.9 Hz, 1H, H-4), 7.604 (dd, J = 8.6 Hz, 2.1 Hz, 1H, H-7), 7.576 (d, J = 2.1 Hz, 1H, H-5), 7.267 (d, J = 8.5 Hz, 1H, H-8), 4.323–4.476 (m, 4H, two CH_2_O), 1.412 (t, J = 7.1 Hz, 3H, CH_3_CH_2_OP), 1.391 (t, J = 7.1 Hz, 3H, CH_3_CH_2_O).

### 3.2. Procedure for Suzuki Reaction—Preliminary Study 

#### 3.2.1. Procedure for Suzuki Reaction with PEPPSI, Pd(PPh_3_)_4_, and IMesPd(dmba)Cl 

In a Schlenk tube 0.108 g (0.3 mmol, 1 equiv) of diethyl (6-bromo-2-oxo-2*H*-chromen-3-yl)phosphonate **5**, phenylboronic acid (0.044 g, 0.36 mmol, 1.2 equiv), finely powdered K_2_CO_3_ (0.124 g, 0.9 mmol, 3 equiv) were placed. A mixture of water:toluene (0.6 mL:2.4 mL), toluene (3 mL), water:dioxane (0.6 mL:2.4 mL), dioxane (3 mL), ethanol:water:toluene (0.5 mL:1 mL:1 mL), or ethanol:water (2 mL:1 mL) were added. The vessel was evacuated under vacuum and refilled with argon four times. After that, the respective catalyst (PEPPSI-type, Pd(PPh_3_)_4_, and IMesPd(dmba)Cl) (3 mol %) was introduced as 0.5 mL THF solution. The resulting suspension was heated at 80 °C for 20 h. After that, the reaction mixture was cooled to room temperature, diluted with CHCl_3_, dried with Na_2_SO_4_ if water was used, and filtered through a celite pad. Part of the solution (about 3 mL) was evaporated under reduced pressure; the residue was dissolved in CDCl_3_ and was analyzed by NMR spectroscopy.

#### 3.2.2. Procedure for Suzuki Reaction with Pd(PPh_3_)_2_Cl_2_


In a Schlenk tube, 0.108 g (0.3 mmol, 1 equiv) of diethyl (6-bromo-2-oxo-2*H*-chromen-3-yl)phosphonate **5**, phenylboronic acid (0.044 g, 0.36 mmol, 1.2 equiv), finely powdered K_2_CO_3_ (0.124 g, 0.9 mmol, 3 equiv) were placed. A mixture of water:toluene (0.6 mL:2.4 mL), toluene (3 mL), water:dioxane (0.6 mL:2.4 mL), dioxane (3 mL), ethanol:water:toluene (0.5 mL:1 mL:1 mL) or ethanol:water (2 mL:1 mL) were added. After that, the catalyst Pd(PPh_3_)_2_Cl_2_ was introduced as solid, and 0.5 mL of THF were additionally added. The vessel was evacuated under vacuum and refilled with argon four times. The resulting suspension was heated at 80 °C for 20 h. After that, the reaction mixture was cooled to room temperature, diluted with CHCl_3_, dried with Na_2_SO_4_ if water was used, and filtered through a celite pad. Part of the solution (about 3 mL) was evaporated under reduced pressure. The residue was dissolved in CDCl_3_ and was analyzed by NMR spectroscopy.

### 3.3. General Procedure for the Preparative Suzuki Reaction 

In a 50 mL Schlenk flask, a mixture of the respective brominated heterocycle 1.5 mmol (1 equiv), boronic acid 1.8 mmol (1.2 equiv), potassium carbonate 0.622 g (4.5 mmol, 3 equiv), and palladium complex 45 µmol (3 mol%)–29 mg PEPPSI or 32 mg Pd(PPh_3_)_2_Cl_2_, was placed, and the respective solvent was added—3 mL:12 mL water:toluene when a PEPPSI-type catalyst was used, or 15 mL toluene for Pd(PPh_3_)_2_Cl_2_. The flask was evacuated and refilled with argon (four times). The reaction was heated at 80 ^o^C until the coumarin derivative was consumed (TLC-monitoring). The needed time for the reactions is given in Table 2, Table 3 and Table 4. The reaction was quenched by adding water to the reaction, and the organic layer was extracted with chloroform (5 × 20 mL). The organic extracts were dried with anhydrous sodium sulfate. After the evaporation of the solvent, the residue was purified by column chromatography using *n*-hexane/EtOAc, *n*-hexane/CH_2_Cl_2_, or *n*-hexane/Et_2_O as gradient eluent system.


**
*Diethyl (2-oxo-6-phenyl-2H-chromen-3-yl)phosphonate, 3a*
**


The product was prepared from diethyl (6-bromo-2-oxo-2*H*-chromen-3-yl)phosphonate **5** (0.540 g), phenylboronic acid (0.220 g), K_2_CO_3_ (0.622 g) and PEPPSI (29 mg) in 3 mL:12 mL water:toluene for 17 h. The product was purified by column chromatography using *n*-hexane/Et_2_O, 0.482 g, 90%, white powder, m.p. = 146.3–149.3 °C. IR (nujol): ν = 1738, 1240, 1052, 1037 cm^−1^.



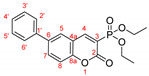



^1^H NMR (500 MHz, CDCl_3_) δ = 8.509 (d, *J* = 17.1 Hz, 1H, H-4), 7.780 (dd, *J* = 8.6 Hz, 2.1 Hz, 1H, H-7), 7.690 (d, *J* = 2.1 Hz, 1H, H-5), 7.498 (m, 2H, H-4′, H-8), 7.413 (m as t, 2H, H-2′ and H-6′), 7.337 (m as q, 2H, H-3′and H-5′), 4.153–4.283 (m, 4H, two CH_3_CH_2_O), 1.330 (t, *J* = 7.1 Hz, 6H, two CH_3_CH_2_O);^13^C NMR (125.7 MHz, CDCl_3_) δ = 158.24 (d, *J_CP_* = 15.1 Hz, C-2), 154.59 (s, C-8a), 153.46 (d, *J_CP_* = 6.5 Hz, C-4), 138.91 (s, C-6), 138.38 (s, C-1′), 133.14 (s, C-7), 129.15 (s, C-2′ and C-6′), 128.08 (s, C-5), 127.33 (C-4′), 127.03 (s, C-3′ and C-5′), 118.27 (d, *J_CP_* = 196.2 Hz, C-3), 118.18 (d, *J_CP_* = 14.3 Hz, C-4a), 117.29 (s, C-8), 63.46 (d, *J_CP_* = 6.0 Hz, two CH_3_CH_2_O), 16.41 (d, *J_CP_* = 6.0 Hz, two CH_3_CH_2_O); ^31^P NMR (161.98 MHz, CDCl_3_) δ = 10.765.

HRMS (ESI) *m*/*z* calculated for C_19_H_19_O_5_P [M+H]^+^ 359.10429 found 359.10413 (ppm: 0.45).


**
*Diethyl (6-(2-methoxyphenyl)-2-oxo-2H-chromen-3-yl)phosphonate, 3b*
**


The product was prepared from diethyl (6-bromo-2-oxo-2*H*-chromen-3-yl)phosphonate **5** (0.540 g), 2-methoxyphenylboronic acid (0.273 g), K_2_CO_3_ (0.622 g) and Pd(PPh_3_)_2_Cl_2_ (32 mg) in 15 mL toluene for 48 h. The product was purified by column chromatography using *n*-hexane/Et_2_O, 0.366 g, 63%, yellow slowly solidifying oil. IR (nujol): ν = 1739, 1243, 1056, 1024 cm^−1^.



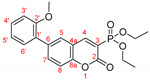



^1^H NMR (400 MHz, CDCl_3_) δ = 8.556 (d, *J* = 17.2 Hz, 1H, H-4), 7.805 (dd, *J* = 8.6 Hz, 2.2 Hz, 1H, H-7), 7.760 (d, *J* = 2.0 Hz, 1H, H-5), 7.392 (d, *J* = 8.5 Hz, 1H, H-8), 7.372 (dd, *J* = 7.5 Hz, 1.1 Hz, 1H, H-3′), 7.308 (dd, *J* = 5.6 Hz, 1.7 Hz, 1H, H-5′), 7.065 (dd*, J* = 7.5 Hz, 1.1 Hz, 1H, H-4′), 7.020 (dd, *J* = 8.3 Hz, 0.7 Hz, 1H, H-6′), 4.203–4.358 (m, 4H, two CH_3_CH_2_O), 3.835 (s, 3H, CH_3_O), 1.391 (t, *J* = 7.1 Hz, 6H, two CH_3_CH_2_O);^13^C NMR (100 MHz, CDCl_3_) δ = 158.49 (d, *J_C_*_P_ = 14.4 Hz, C-2), 156.35 (s, C-8a), 154.25 (s, C-1′), 153.99 (d, *J_CP_* = 6.7 Hz, C-4), 135.77 (s, C-7), 135.54 (s, C-2′), 130.57 (s, C-3′), 129.99 (s, C-5′), 129.55 (s, C-5), 128.14 (s, C-6), 121.10 (C-4′), 117.62 (d, *J_CP_* = 14.3 Hz, C-4a), 117.51 (d, *J_CP_* = 196.2 Hz, C-3), 116.43 (s, C-8), 111.34 (s, C-6′) 63.43 (d, *J_CP_* = 6.2 Hz, two CH_3_CH_2_O), 55.54 (s, CH_3_O), 16.39 (d, *J_CP_* = 6.0 Hz, two CH_3_CH_2_O); ^31^P NMR (161.98 MHz, CDCl_3_) δ = 11.141.

HRMS (ESI) m/z calculated for C_20_H_21_O_6_P [M+H]^+^ 389.11485 found 389.11511 (ppm: −0.67).


**
*Diethyl (6-(4-methoxyphenyl)-2-oxo-2H-chromen-3-yl)phosphonate, 3c*
**


The product was prepared from diethyl (6-bromo-2-oxo-2*H*-chromen-3-yl)phosphonate **5** (0.540 g), 4-methoxyphenylboronic acid (0.273 g), K_2_CO_3_ (0.622 g) and PEPPSI (29 mg) in 3 mL:12 mL water:toluene for 48 h. The product was purified by column chromatography using CH_2_Cl_2_/Et_2_O, 0.551 g, 95%, pale yellow powder, m.p. = 111.6–112.5 °C IR (nujol): ν = 1740, 1242, 1047, 1025 cm^−1^.



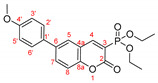



^1^H NMR (400 MHz, CDCl_3_) δ = 8.494 (d, *J* = 17.2 Hz, 1H, H-4), 7.739 (dd, *J* = 8.6 Hz, 2.2 Hz, 1H, H-7), 7.634 (d, *J* = 2.2 Hz, 1H, H-5), 7.328 (d*, J* = 8.6 Hz, 1H, H-8), 7.429 (dd, *J* = 8.8 Hz, 3.1 Hz, 2H, H-3′ and H-5′), 6.937 (dd*, J* = 8.8 Hz, 3.7 Hz, 2H, H-2′ and H-6′), 4.137–4.290 (m, 4H, two CH_3_CH_2_O), 3.793 (s, 3H, CH_3_O), 1.321 (t, *J* = 7.1 Hz, 3H, CH_3_CH_2_O) 1.319 (t, *J* = 7.1 Hz, 3H, CH_3_CH_2_O);^13^C NMR (100 MHz, CDCl_3_) δ = 159.73 (C-4′), 158.31 (d, *J_CP_* = 14.4 Hz, C-2), 154.19 (s, C-8a), 153.54 (d, *J_CP_* = 6.5 Hz, C-4), 138.03 (s, C-6), 132.78 (s, C-7), 131.35 (s, C-1′), 128.11 (s, C-6′and C-2′), 126.71 (s, C-5), 118.16 (d, *J_CP_* = 14.3 Hz, C-4a), 118.10 (d, *J_CP_* = 196.2 Hz, C-3), 117.19 (s, C-8), 114.57 (s, C-3′ and C-5′), 63.44 (d, *J_CP_* = 6.0 Hz, two CH_3_CH_2_O), 55.42 (s, CH_3_O), 16.40 (d*, J_CP_* = 6.2 Hz, two CH_3_CH_2_O); ^31^P NMR (161.98 MHz, CDCl_3_) δ = 10.879.

HRMS (ESI) *m*/*z* calculated for C_20_H_21_O_6_P [M+H]^+^ 389.11485 found 389.11494 (ppm: −0.23).


**
*Diethyl (6-(4-fluorophenyl)-2-oxo-2H-chromen-3-yl)phosphonate, 3d*
**


The product was prepared from diethyl (6-bromo-2-oxo-2*H*-chromen-3-yl)phosphonate **5** (0.540 g), 4-fluorophenylboronic acid (0.251 g), K_2_CO_3_ (0.622 g) and PEPPSI (29 mg) in 3 mL:12 mL water:toluene for 48 h. The product was purified by column chromatography using CH_2_Cl_2_, 0.343 g, 61%, pale yellow powder, m.p. = 115.0–118.2 °C IR (nujol): ν = 1734, 1251, 1057, 1027 cm^−1^.



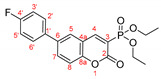



^1^H NMR (400 MHz, CDCl_3_) δ = 8.495 (d, *J* = 17.2 Hz, 1H, H-4), 7.732 (dd, *J* = 8.6 Hz, 2.2 Hz, 1H, H-7), 7.641 (d*, J* = 2.2 Hz, 1H, H-5), 7.466 (ddq, *J_HF_* = 8.8 Hz, 5.0 Hz, *J_HP_* = 8.8 Hz, 3.1 Hz, 2H, H-3′ and H-5′), 7.356 (d, *J* = 8.6 Hz, 1H, H-8), 7.102 (ddq, *J_HF_* = 8.6 Hz, 3.1 Hz, *J_HP_* = 8.6 Hz, 3.1 Hz, 2H, H-2′ and H-6′), 4.143–4.296 (m, 4H, two CH_3_CH_2_O), 1.324 (t, *J* = 7.1 Hz, 3H, CH_3_CH_2_O) 1.323 (t, *J* = 7.1 Hz, 3H, CH_3_CH_2_O);^13^C NMR (100 MHz, CDCl_3_) δ = 162.84 (d, *J_CF_* = 248.0 Hz, C-4′), 158.16 (d, *J_CP_* = 14.4 Hz, C-2), 154.54 (s, C-8a), 153.26 (d, *J_CP_* = 6.5 Hz, C-4), 137.39 (s, C-6), 135.07 (d, *J_CP_* = 3.3 Hz, C-1′), 132.94 (s, C-7), 128.71 (d, *J_CP_* = 10.5 Hz, C-6′and C-2′), 127.17 (s, C-5), 118.50 (d, *J_CP_* = 196.7 Hz, C-3), 118.21 (d, *J_CP_* = 15.5 Hz, C-4a), 117.36 (s, C-8), 116.10 (d, *J_CP_* = 21.7 Hz, C-3′ and C-5′), 63.47 (d, *J_CP_* = 6.0 Hz, two CH_3_CH_2_O), 16.39 (d, *J_CP_* = 6.0 Hz, two CH_3_CH_2_O); ^19^F NMR (376.46 MHz, CDCl_3_) δ = -114.18; ^31^P NMR (161.98 MHz, CDCl_3_) δ = 10.637.

HRMS (ESI) *m*/*z* calculated for C_19_H_18_FO_5_P [M+H]^+^ 377.09486 found 377.09473 (ppm: 0.34).


**
*Diethyl (6-mesityl-2-oxo-2H-chromen-3-yl)phosphonate, 3e*
**


The product was prepared from diethyl (6-bromo-2-oxo-2*H*-chromen-3-yl)phosphonate **5** (0.540 g), 2,4,6-trimethylphenylboronic acid (0.294 g), K_2_CO_3_ (0.622 g) and Pd(PPh_3_)_2_Cl_2_ (32 mg) in 15 mL toluene for 48 h. The product was purified by column chromatography using *n*-hexane/Et_2_O, 0.270 g, 45%, white powder, m.p. = 130.0–132.6 °C. IR (nujol): ν = 1744, 1242, 1053, 1013 cm^−1^.



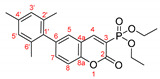



^1^H NMR (400 MHz, CDCl_3_) δ = 8.535 (d, *J* = 17.2 Hz, 1H, H-4), 7.428(s, 1H, H-5), 7.425 (s, 1H, H-8), 7.376 (dd as t, *J* = 1.5 Hz, 1H, H-7), 6.972 (d*, J_HP_* = 0.4 Hz, 2H, H-3′ and H-5′), 4.215–4.367 (m, 4H, two CH_3_CH_2_O), 2.345 (s, 3H, CH_3_), 1.993 (s, 6H, two CH_3_), 1.397 (t, *J* = 7.1 Hz, 6H, two CH_3_CH_2_O);^13^C NMR (100 MHz, CDCl_3_) δ = 158.34 (d, *J_CP_* = 14.2 Hz, C-2), 154.12 (s, C-8a), 153.58 (d, *J_CP_* = 6.7 Hz, C-4), 138.10 (s, C-6), 137.55 (s, C-5), 136.49 (s, C-1′), 135.89 (s, C-6′and C-2′), 135.70 (d, *J_CF_* = 248.0 Hz, C-4′), 129.74 (s, C-7), 128.40 (d, *J_CP_* = 21.7 Hz, C-3′ and C-5′), 118.02 (d, *J_CP_* = 14.3 Hz, C-4a), 118.01 (d, *J_CP_* = 196.1 Hz, C-3),116.96 (s, C-8), 63.44 (d, *J_CP_* = 5.9 Hz, two CH_3_CH_2_O), 21.03 (s, CH_3_), 20.76 (s, two CH_3_), 16.40 (d, *J_CP_* = 6.3 Hz, two CH_3_CH_2_O); ^31^P NMR (161.98 MHz, CDCl_3_) δ = 10.984.

HRMS (ESI) *m*/*z* calculated for C_22_H_25_O_5_P [M+H]^+^ 401.15124 found 401.15143 (ppm: −0.47).


**
*Ethyl 2-oxo-6-phenyl-2H-chromene-3-carboxylate, 9a*
**


The product was prepared from ethyl 6-bromo-2-oxo-2*H*-chromene-3-carboxylate **8a** (0.444 g), phenylboronic acid (0.219 g), K_2_CO_3_ (0.622 g) and PEPPSI (29 mg) in 3 mL:12 mL water:toluene for 30 min. The product was purified by column chromatography using *n*-hexane/Et_2_O and then CH_2_Cl_2_, 0.397 g, 90%, pale yellow powder, m.p. = 150.4–151.5 °C. IR (nujol): ν = 1758, 1694 cm^−1^.



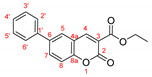



^1^H NMR (400 MHz, CDCl_3_) δ = 8.510 (s, 1H, H-4), 7.783 (dd, *J* = 10.8 Hz, 6.4 Hz, 1H, H-7), 7.709 (d, *J* = 2.2 Hz, 1H, H-5), 7.503 (dd as dt, *J* = 7 Hz, 1.5 Hz, 2H, H-2′ and H-6′), 7.411 (ddd as dt, *J* = 7.4 Hz, 1.5 Hz, 2H, H-3′and H-5′), 7.352 (d, *J* = 8.6 Hz, 1H, H-8), 7.326 (tt, *J* = 7.2 Hz, 1.3 Hz, 1H, H-4′), 4.356 (q, *J* = 7.1 Hz, 2H, CH_3_CH_2_O), 1.348 (t, *J* = 7.1 Hz, 3H, CH_3_CH_2_O); ^13^C NMR (100 MHz, CDCl_3_) δ = 163.06 (s, COOEt), 156.69 (s, C-2), 154.49 (s, C-8a), 148.65 (s, C-4), 138.92 (s, C-6), 138.29 (s, C-1′), 133.29 (s, C-7), 129.14 (s, C-3′ and C-5′), 128.07 (s, C-4′), 127.47 (s, C-5), 127.04 (s, C-2′ and C-6′), 118.66 (s, C-3), 118.12 (s, C-4a), 117.19 (s, C-8), 62.05 (s, CH_3_CH_2_O), 14.26 (s, CH_3_CH_2_O).

HRMS (ESI) *m*/*z* calculated for C_18_H_14_O_4_ [M+H]^+^ 295.09649 found 295.09616 (ppm: 1.12).


**
*Ethyl 6-(2-methoxyphenyl)-2-oxo-2H-chromene-3-carboxylate, 9b*
**


The product was prepared from ethyl 6-bromo-2-oxo-2*H*-chromene-3-carboxylate **8a** (0.444 g), 2-methoxyphenylboronic acid (0.273 g), K_2_CO_3_ (0.622 g) and Pd(PPh_3_)_2_Cl_2_ (32 mg) in 15 mL toluene for 24 h. The product was purified by column chromatography using *n*-hexane/CH_2_Cl_2_ and then CH_2_Cl_2_, 0.331 g, 68%, yellow powder, m.p. = 120.0–124.2 °C. IR (nujol): ν = 1773, 1712 cm^−1^.



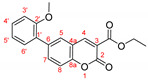



^1^H NMR (400 MHz, CDCl_3_) δ = 8.566 (s, 1H, H-4), 7.811 (dd, *J* = 8.6 Hz, 2.2 Hz, 1H, H-7), 7.757 (d, *J* = 2.1 Hz, 1H, H-5), 7.387 (d, *J* = 8.6 Hz, 1H, H-8), 7.374 (td, *J* = 5.6 Hz, 1.7 Hz, 1H, H-5′), 7.313 (td, *J* = 7.5 Hz, 1.7 Hz, 1H, H-3′), 7.064 (td, *J* = 7.5 Hz, 1.1 Hz, 1H, H-4′), 7.020(dd, *J* = 8.3 Hz, 0.8 Hz, 1H, H-6′), 4.418 (q, *J* = 7.1 Hz, 2H, CH_3_CH_2_O), 3.834 (s, 3H, CH_3_O), 1.421 (t, *J* = 7.1 Hz, 3H, CH_3_CH_2_O); ^13^C NMR (100 MHz, CDCl_3_) δ = 163.20 (s, COOEt), 156.91 (s, C-2), 156.33 (s, C-1′), 154.15 (s, C-8a), 148.99 (s, C-4), 135.93 (s, C-7), 135.48 (s, C-2′), 130.59 (s, C-3′), 130.05 (s, C-5), 129.54 (s, C-5′), 128.20 (s, C-6), 121.10 (s, C-4′), 118.16 (s, C-3), 118.12 (s, C-4a), 117.58 (s, C-8), 111.34 (s, C-6′), 61.98 (s, CH_3_CH_2_O), 55.56 (s, CH_3_O), 14.26 (s, CH_3_CH_2_O).

HRMS (ESI) *m*/*z* calculated for C_19_H_16_O_5_ [M+H]^+^ 325.10705 found 325.1069 (ppm: 0.46).


**
*Ethyl 6-(4-methoxyphenyl)-2-oxo-2H-chromene-3-carboxylate, 9c*
**


The product was prepared from ethyl 6-bromo-2-oxo-2*H*-chromene-3-carboxylate **8a** (0.444 g), 4-methoxyphenylboronic acid (0.273 g), K_2_CO_3_ (0.622 g) and PEPPSI (29 mg) in 3 mL:12 mL water:toluene for 45 min. The product was purified by column chromatography using *n*-hexane/CH_2_Cl_2_ and then CH_2_Cl_2_, 0.444 g, 92%, yellow powder, m.p. = 144.8–146.2 °C. IR (nujol): ν = 1761, 1689 cm^−1^.



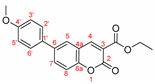



^1^H NMR (400 MHz, CDCl_3_) δ = 8.572 (s, 1H, H-4), 7.816 (dd, *J* = 8.6 Hz, 2.2 Hz, 1H, H-7), 7.726 (d, *J* = 2.2 Hz, 1H, H-5), 7.398 (d, *J* = 9.3 Hz, 1H, H-8), 7.507 (dq, *J* = 8.8 Hz, 3.1 Hz, 2H, H-3′ and H-5′), 7.009 (dq, *J* = 8.8 Hz, 3.1 Hz, 2H, H-4′ and H-6′), 4.419 (q, *J* = 7.1 Hz, 2H, CH_3_CH_2_O), 3.867 (s, 3H, CH_3_O), 1.422 (t, *J* = 7.2 Hz, 3H, CH_3_CH_2_O); ^13^C NMR (100 MHz, CDCl_3_) δ = 163.11 (s, COOEt), 159.72 (s, C-4′), 156.77 (s, C-2), 154.10 (s, C-8a), 148.73 (s, C-4), 137.95 (s, C-6), 132.93 (s, C-7), 131.36 (s, C-1′), 128.10 (s, C-3′ and C-5′), 126.83 (s, C-5), 118.54 (s, C-3), 118.10 (s, C-4a), 117.09 (s, C-8), 114.56 (s, C-2′ and C-6′), 62.02 (s, CH_3_CH_2_O), 55.42 (s, CH_3_O), 14.25 (s, CH_3_CH_2_O).

HRMS (ESI) *m*/*z* calculated for C_19_H_16_O_5_ [M+H]^+^ 325.10705 found 325.10675 (ppm: 0.92).


**
*Ethyl 6-(4-fluorophenyl)-2-oxo-2H-chromene-3-carboxylate, 9d*
**


The product was prepared from ethyl 6-bromo-2-oxo-2*H*-chromene-3-carboxylate **8a** (0.444 g), 4-fluorophenylboronic acid (0.251 g), K_2_CO_3_ (0.622 g) and PEPPSI (29 mg) in 3 mL:12 mL water:toluene for 45 min. The product was purified by column chromatography using CH_2_Cl_2_, 0.374 g, 80%, pale yellow powder, m.p. = 180.3–184.7 °C. IR (nujol): ν = 1742, 1711 cm^−1^.



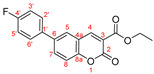



^1^H NMR (400 MHz, CDCl_3_) δ = 8.577 (s, 1H, H-4), 7.808 (dd, *J* = 8.6 Hz, 2.2 Hz, 1H, H-7), 7.738 (d, *J* = 2.2 Hz, 1H, H-5), 7.543 (ddq, *J_HF_* = 8.9 Hz, 5.2 Hz; *J* = 8.9 Hz, 3.2 Hz, 2H, H-3′ and H-5′), 7.423 (d, *J* = 8.6 Hz, 1H, H-8), 7.185 (ddq as tq, *J_HF_* = 8.6 Hz, 3.1 Hz; *J* = 8.6 Hz, 3.1 Hz, 2H, H-2′ and H-6′), 4.432 (q, *J* = 7.1 Hz, 2H, CH_3_CH_2_O), 1.423 (t, *J* = 7.1 Hz, 3H, CH_3_CH_2_O); ^13^C NMR (100 MHz, CDCl_3_) δ = 163.02 (s, COOEt), 162.84 (d, *J* = 248.0 Hz, C-4′), 156.60 (s, C-2), 154.45 (s, C-8a), 148.50 (s, C-4), 137.32 (s, C-6), 135.09 (d, J = 3.2 Hz, C-1′), 133.10 (s, C-7), 128.71 (d, *J* = 8.2 Hz, C-2′ and C-6′), 127.32 (s, C-5), 118.81 (s, C-3), 118.14 (s, C-4a), 117.27 (s, C-8), 116.20 (d, *J* = 21.7 Hz, C-3′ and C-5′), 62.09 (s, CH_3_CH_2_O), 14.24 (s, CH_3_CH_2_O); ^19^F NMR (376.46 MHz, CDCl_3_) δ = -114.196.

HRMS (ESI) *m*/*z* calculated for C_18_H_13_FO_4_ [M+H]^+^ 313.08706 found 313.08682 (ppm: 0.77).


**
*Ethyl 6-mesityl-2-oxo-2H-chromene-3-carboxylate, 9e*
**


The product was prepared from ethyl 6-bromo-2-oxo-2*H*-chromene-3-carboxylate **8a** (0.444 g), 2,4,6-trimethylphenylboronic acid (0.294 g), K_2_CO_3_ (0.622 g) and Pd(PPh_3_)_2_Cl_2_ (32 mg) in 15 mL toluene for 72 h. The product was purified by column chromatography using *n*-hexane/Et_2_O, 0.147 g, 29%, pale yellow powder, m.p. = 142.8–146.2 °C. IR (nujol): ν = 1758, 1691 cm^−1^.



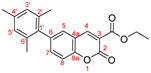



^1^H NMR (400 MHz, CDCl_3_) δ = 8.518 (s, 1H, H-4), 7.427 (s, 1H, H-8), 7.424 (s, 1H, H-5), 7.386 (dd as t, *J* = 1.2 Hz, 1H, H-7), 6.968 (d, *J* = 0.4 Hz, 1H, 2H, H-3′ and H-5′), 4.420 (q, *J* = 7.1 Hz, 2H, CH_3_CH_2_O), 2.343 (s, 3H, CH_3_Ar), 1.993 (s, 6H, CH_3_Ar), 1.416 (t, *J* = 7.1 Hz, 3H, CH_3_CH_2_O); ^13^C NMR (100 MHz, CDCl_3_) 163.16 (s, COOEt), 156.84 (s, C-2), 154.01 (s, C-8a), 148.63 (s, C-4), 138.04 (s, C-6), 137.54 (s, C-5), 136.51 (s, C-1′), 135.91 (s, C-2′ and C-6′), 135.79 (s, C-4′), 129.83 (s, C-7), 128.38 (s, C-3′ and C-5′),118.51 (s, C-3), 117.99 (s, C-4a), 116.95 (s, C-8), 62.02 (s, CH_3_CH_2_O), 21.04 (s, CH_3_Ar), 20.76 (s, two CH_3_Ar), 14.26 (s, CH_3_CH_2_O).

HRMS (ESI) *m*/*z* calculated for C_21_H_20_O_4_ [M+H]^+^ 337.14344 found 337.14352 (ppm: −0.24).


**
*Methyl 2-oxo-6-phenyl-2H-chromene-3-carboxylate, 10a*
**


The product was prepared from methyl 6-bromo-2-oxo-2*H*-chromene-3-carboxylate **8b** (0.425 g), phenylboronic acid (0.219 g), K_2_CO_3_ (0.622 g) and PEPPSI (29 mg) in 3 mL:12 mL water:toluene for 60 min. The product was purified by column chromatography using *n*-hexane/CH_2_Cl_2_ and then CH_2_Cl_2_, 0.409 g, 97%, pale yellow powder, m.p. = 160.7–164.5 °C. IR (nujol): ν = 1755, 1694 cm^−1^.



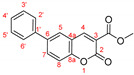



^1^H NMR (400 MHz, CDCl_3_) δ = 8.627 (s, 1H, H-4), 7.871 (dd, *J* = 8.6 Hz, 2.2 Hz, 1H, H-7), 7.785 (d, *J* = 2.2 Hz, 1H, H-5), 7.565–7.589 (m, 2H, H-2′ and H-6′), 7.468–7.487 (m, *J* = 7.4 Hz, 2H, H-3′and H-5′), 7.390–7.432 (m, 1H, H-4′), 7.434 (d, *J* = 8.8 Hz, 1H, H-8), 3.972 (s, *J* = 3H, CH_3_OOC); ^13^C NMR (100 MHz, CDCl_3_) 163.72 (s, COOCH_3_), 156.69 (s, C-2), 154.54 (s, C-8a), 149.22 (s, C-4), 138.87 (s, C-6), 138.37 (s, C-1′), 133.45 (s, C-7), 129.15 (s, C-3′ and C-5′), 128.09 (s, C-4′), 127.52 (s, C-5), 127.04 (s, C-2′ and C-6′), 118.26 (s, C-3), 118.09 (s, C-4a), 117.20 (s, C-8), 52.98 (s, CH_3_O).

HRMS (ESI) *m*/*z* calculated for C_17_H_12_O_4_ [M+H]^+^ 281.08084 found 281.08112 (ppm: −1.00).


**
*Methyl 6-(2-methoxyphenyl)-2-oxo-2H-chromene-3-carboxylate, 10b*
**


The product was prepared from methyl 6-bromo-2-oxo-2*H*-chromene-3-carboxylate **8b** (0.425 g), 2-methoxyphenylboronic acid (0.273 g), K_2_CO_3_ (0.622 g) and Pd(PPh_3_)_2_Cl_2_ (32 mg) in 15 mL toluene for 21 h. The product was purified by column chromatography using *n*-hexane/CH_2_Cl_2_ and then CH_2_Cl_2_, 0.385 g, 83%, yellow powder, m.p. = 130.9–135.5 °C. IR (nujol): ν = 1760, 1759 cm^−1^.



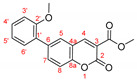



^1^H NMR (400 MHz, CDCl_3_) δ = 8.611 (s, 1H, H-4), 7.819 (dd, *J* = 8.6 Hz, 2.1 Hz, 1H, H-7), 7.763 (d, *J* = 2.1 Hz, 1H, H-5), 7.393 (d, *J* = 8.8 Hz, 1H, H-8), 7.374 (td, *J* = 5.6 Hz, 1.7 Hz 1H, H-5′), 7.311 (dd, *J* = 7.5 Hz, 1.8 Hz, 1H, H-3′), 7.066 (td, *J* = 7.5 Hz, 1.0 Hz, H-4′), 7.022 (d, *J* = 8.3 Hz, H-6′), 3.967 (s, 3H, CH_3_OOC), 3.834 (s, 3H, CH_3_O); ^13^C NMR (100 MHz, CDCl_3_) δ = 163.86 (s, COOCH_3_), 156.91 (s, C-2), 154.21 (s, C-8a), 156.32 (s, C-1′), 149.56 (s, C-4), 136.10 (s, C-7), 135.56 (s, C-2′), 130.59 (s, C-5′), 130.11 (s, C-5),129.57 (s, C-3′), 128.15 (s, C-6), 121.11 (s, C-4′), 117.77 (s, C-3), 117.56 (s, C-4a), 116.33 (s, C-8), 111.34 (s, C-6′), 52.93 (s, CH_3_O).

HRMS (ESI) *m*/*z* calculated for C_18_H_14_O_5_ [M+H]^+^ 311.0914 found 311.09083 (ppm: −1.00).


**
*Methyl 6-(4-methoxyphenyl)-2-oxo-2H-chromene-3-carboxylate, 10c*
**


The product was prepared from methyl 6-bromo-2-oxo-2*H*-chromene-3-carboxylate **8b** (0.425 g), 4-methoxyphenylboronic acid (0.273 g), K_2_CO_3_ (0.622 g) and PEPPSI (29 mg) in 3 mL:12 mL water:toluene for 45 min. The product was purified by column chromatography using *n*-hexane/CH_2_Cl_2_ and then CH_2_Cl_2_, 0.463 g, 99%, lemon yellow powder, m.p. = 158.7–161.4 °C. IR (nujol): ν = 1756, 1700 cm^−1^.



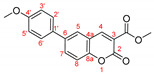



^1^H NMR (400 MHz, CDCl_3_) δ = 8.613 (s, 1H, H-4), 7.826 (dd, *J* = 8.6 Hz, 2.6 Hz, 1H, H-7), 7.726 (d, *J* = 2.2 Hz, 1H, H-5), 7.506 (dq, *J* = 8.6 Hz, 3.0 Hz, 2H, H-3′ and H-5′), 7.403 (d, *J* = 8.6 Hz, 1H, H-8), 7.011 (dq, *J* = 8.8 Hz, 3.0 Hz, 2H, H-2′ and H-6′), 3.969 (s, *J* = 3H, CH_3_OOC), 3.868 (s, *J* = 3H, CH_3_O); ^13^C NMR (100 MHz, CDCl_3_) δ = 163.77 (s, COOCH_3_), 159.75 (s, C-4′), 156.77 (s, C-2), 154.16 (s, C-8a), 149.22 (s, C-4), 138.03 (s, C-6), 133.10 (s, C-7), 131.31 (s, C-1′), 129.15 (s, C-5′ and C-3′), 126.89 (s, C-5), 118.07 (s, C-3), 118.15 (s, C-8), 117.12 (s, C-4a), 114.55 (s, C-2′ and C-6′), 52.96 (s, CH_3_O).

HRMS (ESI) *m*/*z* calculated for C_18_H_14_O_5_ [M+H]^+^ 311.0914 found 311.09106 (ppm: 1.09).


**
*Methyl 6-(4-fluorophenyl)-2-oxo-2H-chromene-3-carboxylate, 10d*
**


The product was prepared from methyl 6-bromo-2-oxo-2*H*-chromene-3-carboxylate **8a** (0.425 g), 4-fluorophenylboronic acid (0.252 g), K_2_CO_3_ (0.622 g) and PEPPSI (29 mg) in 3 mL:12 mL water:toluene for 60 min. The product was purified by column chromatography using CH_2_Cl_2_, 0.443 g, 99%, pale yellow powder, m.p. = 170.8–174.5.7 °C. IR (nujol): ν = 1743, 1712 cm^−1^.



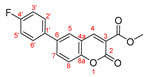



^1^H NMR (400 MHz, CDCl_3_) δ = 8.618 (s, 1H, H-4), 7.819 (dd, *J* = 8.6 Hz, 2.2 Hz, 1H, H-7), 7.734 (d, *J* = 2.2 Hz, 1H, H-5), 7.550 (ddq, *J_HF_* = 8.8 Hz, 5.2 Hz; *J* = 8.8 Hz, 3.1 Hz, 2H, H-3′ and H-5′), 7.430 (d, *J* = 8.6 Hz, 1H, H-8), 7.202 (ddq as tq, *J_HF_* = 8.6 Hz, 3.6 Hz; *J* = 8.6 Hz, 3.2 Hz, 2H, H-2′ and H-6′), 3.973 (s, 3H, CH_3_OOC); ^13^C NMR (100 MHz, CDCl_3_) δ = 163.68 (s, COOEt), 162.86 (d, *J* = 248.0 Hz, C-4′), 156.60 (s, C-2), 154.51 (s, C-8a), 149.07 (s, C-4), 137.40 (s, C-6), 135.05 (d, *J* = 3.3 Hz, C-1′), 133.26 (s, C-7), 128.71 (d, *J* = 10.5 Hz, C-2′ and C-6′), 127.38 (s, C-5), 118.11 (s, C-3), 117.29 (s, C-4a), 118.42 (s, C-8), 116.12 (d, *J* = 21.7 Hz, C-3′ and C-5′), 53.00 (s, CH_3_O); ^19^F NMR (376.46 MHz, CDCl_3_) δ = −114.135.

HRMS (ESI) *m*/*z* calculated for C_17_H_11_FO_4_ [M+H]^+^ 299.07141 found 299.07090 (ppm: 1.71).


*
**Methyl 6-mesityl-2-oxo-2H-chromene-3-carboxylate, 10e**
*


The product was prepared from methyl 6-bromo-2-oxo-2*H*-chromene-3-carboxylate **8a** (0.425 g), 2,4,6-trimethylphenylboronic acid (0.294 g), K_2_CO_3_ (0.622 g) and Pd(PPh_3_)_2_Cl_2_ (32 mg) in 15 mL toluene for 90 h. The product was purified by column chromatography using *n*-hexane/CH_2_Cl_2_, 0.085 g, 18%, white powder, m.p. = 139.9–143.5 °C. IR (nujol): ν = 1745, 1706 cm^−1^.



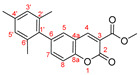



^1^H NMR (400 MHz, CDCl_3_) δ = 8.564 (s, 1H, H-4), 7.438 (s, 1H, H-8), 7.434 (s, 1H, H-5), 7.392 (dd as t, *J* = 1.0 Hz, 1H, H-7), 6.970 (d, *J* = 0.4 Hz, 2H, H-3′ and H-5′), 3.968 (s, 3H, CH_3_OOC), 2.343 (s, 3H, CH_3_Ar), 1.994 (s, 6H, CH_3_Ar); ^13^C NMR (100 MHz, CDCl_3_) 163.80 (s, COOEt), 156.84 (s, C-2), 154.07 (s, C-8a), 149.21 (s, C-4), 138.11 (s, C-6), 137.57 (s, C-5), 136.46 (s, C-1′), 135.97 (s, C-2′ and C-6′), 137.10 (s, C-4′), 129.90 (s, C-7), 128.39 (s, C-3′ and C-5′), 118.57 (s, C-3), 117.96 (s, C-4a), 116.97 (s, C-8), 52.96 (s, CH_3_O), 21.04 (s, CH_3_Ar), 20.77 (s, two CH_3_Ar).

HRMS (ESI) *m*/*z* calculated for C_20_H_18_O_5_ [M+H]^+^ 323.12779 found 323.12753 (ppm: −0.80).


**
*Ethyl 2-ethoxy-6-phenylbenzo[e][1,2]oxaphosphinine-3-carboxylate-2-oxide, 11a*
**


The product was prepared from ethyl 6-bromo-2-ethoxybenzo[e][1,2]oxaphosphinine-3-carboxylate-2-oxide **6** (0.540 g), phenylboronic acid (0.220 g), K_2_CO_3_ (0.622 g) and PEPPSI (29 mg) in 3 mL:12 mL water:toluene for 16 h. The product was purified by column chromatography using *n*-hexane/Et_2_O, 0.476 g, 89%, pale yellow powder, m.p. = 118.7–120.3 °C. IR (nujol): ν = 1715, 1196, 1076, 1028 cm^−1^.



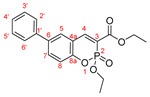



^1^H NMR (400 MHz, CDCl_3_) δ = 8.307 (d, *J* = 36.9 Hz, 1H, H-4), 7.702 (dd, *J* = 8.6 Hz, 2.0 Hz, 1H, H-7), 7.657 (d, *J* = 2.1 Hz, 1H, H-5), 7.545 (dq, *J* = 7.6 Hz, 2.0 Hz, 1.5 Hz, 2H, H-2′ and H-6′), 7.459 (tt, *J* = 2.0 Hz, 1.5 Hz, 2H, H-3′and H-5′), 7.382 (tt, *J* = 7.3, 2.2, 1.3 Hz, 1H, H-4′), 7.266 (d, *J* = 8.5 Hz, 1H, H-8), 4.413–4.493 (m, 2H, CH_3_CH_2_OP), 4.323–4.403 (m, 2H, CH_3_CH_2_OOC), 1.438 (t, *J* = 7.0 Hz, 3H, CH_3_CH_2_OP), 1.403 (t, *J* = 7.1 Hz, 3H, CH_3_CH_2_OOC);^13^C NMR (100 MHz, CDCl_3_) δ = 163.75 (d, *J*_CP_ = 14.7 Hz, COOEt), 152.01 (d, *J*_CP_ = 8.7 Hz, C-8a), 150.46 (d, *J*_CP_ = 3.5 Hz, C-4), 139.06 (s, C-1′), 137.59 (s, C-6), 132.34 (s, C-7), 129.87 (d, *J*_CP_ = 1.5 Hz, C-5), 127.85 (C-4′), 129.04 (s, C-3′ and C-5′), 126.93 (s, C-2′ and C-6′), 119.68 (s, C-4a), 119.19 (d, *J*_CP_ = 7.5 Hz, C-8), 118.88 (d, *J*_CP_ = 193.3 Hz, C-3), 64.93 (d, *J*_CP_ = 6.3 Hz, CH_3_CH_2_OP), 62.04 (s, CH_3_CH_2_O), 16.47 (d, *J*_CP_ = 6.5 Hz, CH_3_CH_2_OP), 14.24 (s, CH_3_CH_2_O); ^31^P NMR (161.98 MHz, CDCl_3_) δ = 5.321.

HRMS (ESI) *m*/*z* calculated for C_19_H_19_O_5_P [M+H]^+^ 359.10429 found 359.10446 (ppm: −0.47).


**
*Ethyl 2-ethoxy-6-(2-methoxyphenyl)benzo[e][1,2]oxaphosphinine-3-carboxylate-2-oxide, 11b*
**


The product was prepared from ethyl 6-bromo-2-ethoxybenzo[e][1,2]oxaphosphinine-3-carboxylate-2-oxide **6** (0.540 g), 2-methoxyphenylboronic acid (0.273 g), K_2_CO_3_ (0.622 g) and Pd(PPh_3_)_2_Cl_2_ (32 mg) in 15 mL toluene for 42 h. The product was purified by column chromatography using *n*-hexane/EtOAc, 0.366 g, 45%, yellow oil. IR (nujol): ν = 1700, 1190, 1074, 1035 cm^−1^.



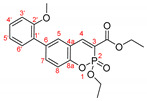



^1^H NMR (400 MHz, CDCl_3_) δ = 8.284 (d, *J* = 37.0 Hz, 1H, H-4), 7.651 (dd, *J* = 8.6 Hz, 2.5 Hz, 1H, H-7), 7.636 (s, 1H, H-5), 7.355 (td, *J* = 8.0 Hz, 2.2 Hz, 1H, H-3′), 7.283 (dd, *J* = 7.5, 1.7 Hz, 1H, H-5′), 7.223 (d, *J* = 8.2 Hz, 1H, H-8), 7.023 (td, *J* = 7.4, 0.9 Hz, 1H, H-6′), 6.982 (td, *J* = 7.5 Hz, 1.1 Hz, 1H, H-4′), 4.315–4.487 (m, 4H, CH_3_CH_2_OP and CH_3_CH_2_OOC), 3.820 (s, 3H, OCH_3_), 1.420 (t, *J* = 7.1 Hz, 3H, CH_3_CH_2_OP), 1.406 (t, *J* = 7.1 Hz, 3H, CH_3_CH_2_OOC); ^13^C NMR (100 MHz, CDCl_3_) δ = 163.89 (d, *J*_CP_ = 13 Hz, COOEt), 156.32 (s, C-1′), 151.63 (d, *J*_CP_ = 8.8 Hz, C-8a), 150.89 (d, *J*_CP_ = 3.6 Hz, C-4), 134.98 (s, C-7), 134.74 (s, C-5), 130.54 (s, C-5′), 129.32 (s, C-3′), 128.36 (s, C-6), 121.04 (s, C-6′), 121.03 (C-4′), 119.18 (d, *J*_CP_ = 15.8 Hz, C-4a), 118.36 (d, *J*_CP_ = 7.4 Hz, C-8), 118.35 (d, *J*_CP_ = 177.7 Hz, C-3), 111.31 (s, C-2′), 64.78 (d, *J*_CP_ = 6.4 Hz, CH_3_CH_2_OP), 61.98 (s, CH_3_CH_2_O), 55.56 (s, OCH_3_), 16.46 (d, *J*_CP_ = 6.5 Hz, CH_3_CH_2_OP), 14.24 (s, CH_3_CH_2_O); ^31^P NMR (161.98 MHz, CDCl_3_) δ = 5.537.

HRMS (ESI) *m*/*z* calculated for C_20_H_21_O_6_P [M+H]^+^ 389.11485 found 389.1149 (ppm: 0.13).


**
*Ethyl 2-ethoxy-6-(4-methoxyphenyl)benzo[e][1,2]oxaphosphinine-3-carboxylate-2-oxide, 11c*
**


The product was prepared from ethyl 6-bromo-2-ethoxybenzo[e][1,2]oxaphosphinine-3-carboxylate-2-oxide **6** (0.540 g), 4-methoxyphenylboronic acid (0.273 g), K_2_CO_3_ (0.622 g) and PEPPSI (29 mg) in 3 mL:12 mL water:toluene for 3 h. The product was purified by column chromatography using *n*-hexane/EtOAc, 0.506 g, 87%, pale yellow powder, m.p. = 132.7–135.3 °C IR (nujol): ν = 1720, 1195, 1043, 1026 cm^−1^.



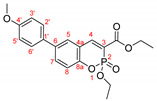



^1^H NMR (400 MHz, CDCl_3_) δ = 8.296 (d, *J* = 36.9 Hz, 1H, H-4), 7.662 (dd, *J* = 8.6 Hz, 2.0 Hz, 1H, H-7), 7.603 (d, *J* = 2.3 Hz, 1H, H-5), 7.475 (dq, *J* = 8.8 Hz, 3.0 Hz, 2H, H-2′ and H-6′), 6.989 (dq, *J* = 8.8 Hz, 3.0 Hz, 2H, H-3′and H-5′), 7.237 (d, *J* = 8.5 Hz, 1H, H-8), 4.319–4.490 (m, 4H, CH_3_CH_2_OP and CH_3_CH_2_OOC), 1.423 (t, *J* = 7.0 Hz, 3H, CH_3_CH_2_OP), 1.409 (t, *J* = 7.1 Hz, 3H, CH_3_CH_2_OOC);^13^C NMR (100 MHz, CDCl_3_) δ = 163.78 (d, *J*_CP_ = 13.0 Hz, COOEt), 151.58 (d, *J*_CP_ = 8.6 Hz, C-8a), 150.59 (d, *J*_CP_ = 3.5 Hz, C-4), 131.55 (s, C-1′), 137.26 (s, C-6), 131.95 (s, C-7), 129.35 (d, *J*_CP_ = 1.2 Hz, C-5), 159.55 (C-4′), 114.47 (s, C-3′ and C-5′), 127.99 (s, C-2′ and C-6′), 119.64 (s, C-4a), 119.11 (d, *J*_CP_ = 7.4 Hz, C-8), 118.66 (d, *J*_CP_ = 177.3 Hz, C-3), 64.89 (d, *J*_CP_ = 6.3 Hz, CH_3_CH_2_OP), 62.02 (s, CH_3_CH_2_O), 16.47 (d, *J*_CP_ = 6.4 Hz, CH_3_CH_2_OP), 14.24 (s, CH_3_CH_2_O); ^31^P NMR (161.98 MHz, CDCl_3_) δ = 5.408.

HRMS (ESI) *m*/*z* calculated for C_20_H_21_O_6_P [M+H]^+^ 389.11485 found 389.11517 (ppm: −0.82).


*
**Ethyl 2-ethoxy-6-(4-fluorophenyl)benzo[e][1,2]oxaphosphinine-3-carboxylate-2-oxide, 11d**
*


The product was prepared from ethyl 6-bromo-2-ethoxybenzo[e][1,2]oxaphosphinine-3-carboxylate-2-oxide **6** (0.540 g), 4-fluorophenylboronic acid (0.251 g), K_2_CO_3_ (0.622 g) and PEPPSI (29 mg) in 3 mL:12 mL water:toluene for 2.5 h. The product was purified by column chromatography using *n*-hexane/EtOAc, 0.500 g, 89%, pale yellow powder, m.p. = 115.0–118.2 °C IR (nujol): ν = 1718, 1198, 1078, 1028 cm^−1^.



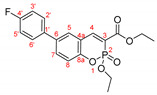



^1^H NMR (400 MHz, CDCl_3_) δ = 8.294 (d, *J* = 36.5 Hz, 1H, H-4), 7.647 (dd, *J* = 8.5 Hz, 2.1 Hz, 1H, H-7), 7.606 (d, *J* = 2.2 Hz, 1H, H-5), 7.502 (ddq, *J* = 8.8 Hz, 5,2 Hz, 3.4 Hz, 2H, H-2′ and H-6′), 7.149 (tq, *J* = 8.6 Hz, 3.4 Hz, 2H, H-3′and H-5′), 7.261 (d, *J* = 8.4 Hz, 1H, H-8), 4.323–4.498 (m, 4H, CH_3_CH_2_OP and CH_3_CH_2_OOC), 1.429 (t, *J* = 7.1 Hz, 3H, CH_3_CH_2_OP), 1.411 (t, *J* = 7.1 Hz, 3H, CH_3_CH_2_OOC);^13^C NMR (100 MHz, CDCl_3_) δ = 163.68 (d, *J*_CP_ = 13.0 Hz, COOEt), 151.98 (d, *J*_CP_ = 8.6 Hz, C-8a), 150.27 (d, *J*_CP_ = 3.5 Hz, C-4), 135.23 (d, *J* = 3.1 Hz C-1′), 136.62 (s, C-6), 132.16 (s, C-7), 129.71 (s, C-5), 162.72 (d, *J*_CP_ = 247.7 Hz, C-4′), 128.58 (d, *J*_CP_ = 8.1 Hz, C-3′ and C-5′), 115.98 (d, *J*_CP_ = 21.6 Hz, C-2′ and C-6′), 119.88 (s, C-4a), 119.27 (d, *J*_CP_ = 7.4 Hz, C-8), 118.92 (d, *J*_CP_ = 162.2 Hz, C-3), 64.98 (d, *J*_CP_ = 6.3 Hz, CH_3_CH_2_OP), 62.07 (s, CH_3_CH_2_O), 16.47 (d, *J*_CP_ = 6.4 Hz, CH_3_CH_2_OP), 14.24 (s, CH_3_CH_2_O); ^31^P NMR (161.98 MHz, CDCl_3_) δ = 5.232; ^19^F NMR (376.46 MHz, CDCl_3_) δ = −114.590.

HRMS (ESI) *m*/*z* calculated for C_19_H_18_FO_5_P [M+H]^+^ 377.09486 found 377.09515 (ppm: −0.77).

### 3.4. General Procedure for the Sonogashira Reaction 

A mixture of the corresponding coumarin derivative (1.4 mmol), aryl ethyne (1.8 mmol, 1.2 equiv.), PdCl_2_(PPh_3_)_2_/PEPPSI (2 mol%), CuI (4 mol%), Et_3_N (3 equiv), and DMF (15 mL) was heated at 80 °C under an Ar atmosphere until the coumarin derivative was consumed (TLC-monitoring). The needed time for the reactions is given in Table 5. Then, it was concentrated under reduced pressure. The crude product was dissolved in EtOAc, watched several times with brine (5 × 5 mL), and once with water (1 × 10 mL). The organic layer was dried with anhydrous sodium. After the evaporation of the solvent, the residue was purified by column chromatography.


**
*Diethyl (2-oxo-6-(phenylethynyl)-2H-chromen-3-yl)phosphonate, 12a*
**


The product was prepared from diethyl (6-bromo-2-oxo-2*H*-chromen-3-yl)phosphonate **5** (0.500 g), ethynylbenzene (0.170 g), PdCl_2_(PPh_3_)_2_ (19.4 mg), CuI (10.5 mg), Et_3_N (0.58 mL, 0.42 g) and DMF (15 mL) for 20 h. The product was purified by column chromatography using *n*-hexane/EtOAc, 0.437 g, 82%, reddish colored powder, m.p. = 94.3–99.6 °C. IR (nujol): ν = 1745, 1239, 1046, 1018 cm^−1^.



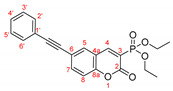



^1^H NMR (500 MHz, CDCl_3_) δ = 8.413 (d, *J* = 17.2 Hz, 1H, H-4), 7.691 (s, 1H, H-5), 7.674 (s, 1H, H-7), 7.465 (s, 1H, H-8), 7.297 (s, 2H, H-2′ and H-6′), 7.271 (s, 2H, H-3′ and H-5′), 7.254 (s, 1H, H-4′), 4.146–4.261 (m, 4H, two CH_3_CH_2_O), 1.316 (t, *J* = 7.1 Hz, 6H, two CH_3_CH_2_O);^13^C NMR (125.7 MHz, CDCl_3_) δ = 157.81 (d, *J*_CP_ = 14.4 Hz, C-2), 154.62 (s, C-8a), 152.64 (d, *J*_CP_ = 6.6 Hz, C-4), 137.06 (s, C-7), 132.07 (s, C-5), 131.67 (s, C-6′and C-2′), 128.83 (s, C-4′), 128.48 (s, C-3′ and C-5′), 122.44 (s, C-6), 120.49 (s, C-1′), 118.75 (d, *J*_CP_ = 196.4 Hz, C-3), 117.99 (d, *J*_CP_ = 14.5 Hz, C-4a), 117.93 (s, C-8), 90.67 (s, C≡C), 87.02 (s, C≡C), 63.57 (d, *J*_CP_ = 6.9 Hz, two CH_3_CH_2_O), 16.38 (d, *J*_CP_ = 6.3 Hz, two CH_3_CH_2_O); ^31^P NMR (161.98 MHz, CDCl_3_) δ = 10.343.

HRMS (ESI) *m*/*z* calculated for C_21_H_19_O_5_P [M+H]^+^ 383.10429 found 383.10468 (ppm: −1.02).


**
*Ethyl 2-oxo-6-(phenylethynyl)-2H-chromene-3-carboxylate, 12b*
**


The product was prepared from ethyl 6-bromo-2-oxo-2H-chromene-3-carboxylate **8a** (0.444 g), ethynylbenzene (0.184 g), PdCl_2_(PPh_3_)_2_ (21 mg), CuI (11.4 mg), Et_3_N (0.63 mL, 0.46 g) and DMF (15 mL) for 22 h. The product was purified by column chromatography using *n*-hexane/EtOAc, 0.247 g, 52%, reddish colored powder, m.p. = 57.1–61.8 °C. IR (nujol): ν = 1744, 1711 cm^−1^.



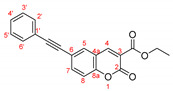



^1^H NMR (400 MHz, CDCl_3_) δ = 8.485 (s, 1H, H-4), 7.774 (s, 1H, H-5), 7.755 (dd, *J* = 7.3 Hz, 2.0 Hz, 1H, H-7), 7.530–7.555 (m, 2H, H-2′ and H-6′), 7.341 (dd, *J* = 9.2 Hz, 0.4 Hz, 1H, H-8), 7.363–7.388 (m, 2H, H-3′ and H-5′), 4.421 (q, *J* = 7.1 Hz, 2H, CH_3_CH_2_O), 1.420 (t, *J* = 7.1 Hz, 3H, CH_3_CH_2_O); ^13^C NMR (100 MHz, CDCl_3_) δ = 162.87 (s, COOEt), 156.23 (s, C-2), 154.54 (s, C-8a), 147.82 (s, C-4), 122.46 (s, C-6), 120.43 (s, C-1′), 137.19 (s, C-7), 128.49 (s, C-3′ and C-5′), 128.83 (s, C-4′), 132.19 (s, C-5), 131.66 (s, C-2′ and C-6′), 117.94 (s, C-3), 119.11 (s, C-4a), 117.10 (s, C-8), 62.16 (s, CH_3_CH_2_O), 14.22 (s, CH_3_CH_2_O).

HRMS (ESI) *m*/*z* calculated for C_20_H_14_O_4_ [M+H]^+^ 319.09649 found 319.09641 (ppm: 0.25).


**
*Diethyl (6-((4-methoxyphenyl)ethynyl)-2-oxo-2H-chromen-3-yl)phosphonate, 12d*
**


The product was prepared from diethyl (6-bromo-2-oxo-2H-chromen-3-yl)phosphonate **5** (0.540 g), 1-ethynyl-4-methoxybenzene (0.238 g), PEPPSI (19.4 mg), CuI (11.4 mg), Et_3_N (0.62 mL, 0.55 g) and DMF (15 mL) for 30 h. The product was purified by column chromatography using *n*-hexane/EtOAc, 0.349 g, 56%, reddish colored powder, m.p. = 131.7–136.0 °C. IR (nujol): ν = 2211, 1744, 1243, 1060, 1027 cm^−1^.



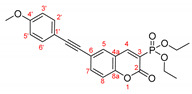



^1^H NMR (400 MHz, CDCl_3_) δ = 8.478 (d, *J* = 17.2 Hz, 1H, H-4), 7.735 (dd, *J* = 8.5 Hz, 2.0 Hz, 1H, H-7), 7.716 (d, *J* = 2.0 Hz, 1H, H-5), 7.327 (d, *J* = 8.5 Hz, 1H, H-8), 7.485 (dq, *J* = 8.8 Hz, 2.8 Hz, 2H, H-2′ and H-6′), 6.901 (dq, *J* = 8.8 Hz, 2.8 Hz, 2H, H-3′ and H-5′), 4.226–4.343 (m, 4H, two CH_3_CH_2_O), 3.844 (s, 3H, CH_3_O), 1.392 (t, *J* = 7.1 Hz, 6H, two CH_3_CH_2_O); ^13^C NMR (100 MHz, CDCl_3_) δ = 160.06 (s, C-4′), 157.82 (s, C-2), 154.43 (s, C-8a), 152.69 (d, J_CP_ = 6.6 Hz, C-4), 136.95 (s, C-7), 131.79 (s, C-5), 133.19 (s, C-6′and C-2′), 114.49 (s, C-3′ and C-5′), 120.90 (s, C-6), 119.69 (s, C-1′), 118.72 (d, *J_CP_* = 196.1 Hz, C-3), 117.98 (d, *J_CP_* = 14.5 Hz, C-4a), 117.13 (s, C-8), 90.79 (s, C≡C), 85.83 (s, C≡C), 55.35 (s, CH_3_O), 63.54 (d, *J_CP_* = 6.0 Hz, two CH_3_CH_2_O), 16.39 (d, *J_CP_* = 6.3 Hz, two CH_3_CH_2_O); ^31^P NMR (161.98 MHz, CDCl_3_) δ = 10.390.

HRMS (ESI) *m*/*z* calculated for C_22_H_21_O_6_P [M+H]^+^ 413.11485 found 413.11524 (ppm: −0.94).

## 4. Conclusions 

A synthetic protocol for the efficient preparation of new substituted 3-phosphonocoumarins, 1,2-benzoxaphosphorines, methyl and ethyl coumarin-3-carboxylates via palladium catalyzed cross-coupling reactions was developed. The Suzuki and Sonogashira reaction was applied successfully to obtain the structures in good to quantitative yields.

Quantum-chemical calculations on the spectral properties of aryl and alkynyl 3-phosphonocoumarins were performed as a continuation of our systematic studies on phosphorous-containing coumarin derivatives. It was found that by altering the substituent in C-6, fine tuning of the fluorescent properties could be achieved. Introducing a CH_3_O-group in *para*-position in the aryl moiety has the most noticeable effect leading to bathochromic shift in the absorption and fluorescence spectra. This tendency was observed both in the calculated and the experimental data. The photophysical properties of the newly obtained compounds were studied in different solvents. It was found that the type of media has little or no effect on both absorption and fluorescence maxima. The experimental data are in good agreement with the theoretically predicted photophysical properties of the compounds.

## Data Availability

Not applicable.

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
