# Peer review of "Efficient Synthesis of Fluorescent Coumarins and Phosphorous-Containing Coumarin-Type Heterocycles via Palladium Catalyzed Cross-Coupling Reactions"

_molecules, 2022, doi:10.3390/molecules27217649_

Round 1

Reviewer 1 Report

The paper "Efficient synthesis of fluorescent coumarins and phosphorous 

containing coumarin type heterocycles via palladium catalyzed 

cross-coupling reactions" is of great interest. The methods are clearly described, the yields obtained are low to really good but reliable. This paper is nice.

I would just have a question to authors : did they try to recycle the palladium media from one or two experiments, if yes, they should include the results, and what was the leaching percentage between the two?

Author Response

We would like to thank the reviewer for the comments on our manuscript.

Point 1: Did they try to recycle the palladium media from one or two experiments, if yes, they should include the results, and what was the leaching percentage between the two?

Answer: We did not try to recycle the used palladium media due to the low amount of the used catalysts. For the experiments overall around 1 g of PEPPSI was used. On one hand, the content of palladium in the complex is 16%, therefore where are around 160 mg of elemental palladium in it.  On the other hand, the palladium catalyst could transform itself into several forms: elemental palladium or different not active complexes, these forms complicate its collecting and further recycling.

Reviewer 2 Report

The paper by Koleva et al. describes a predictive chemistry approach to the development of fluorescent 6-aryl-3-phosphonocoumarins with “push-pull” effect. DFT and (TD)DFT calculation were used to predict the photophysical properties of a series of compounds, and the data was used to target the synthesis of products with good spectral properties. The paper is very well written. The reactions developed are not new but are well optimized. Overall, the novelty of the project is not high but the approach and results are interesting, therefore I recommend publication in molecules once the following points have been addressed.

-          It is not clear how the authors have selected the substituents for the model compounds in the DFT study. A more systematic analysis would have been possible by a more judicious design. The authors should evaluate the CN in para and also the OMe in meta to have a better picture of the effects.

-          The authors optimize the Suzuki reaction by testing different catalysts and solvents. Do the authors use an internal standard? If this is the case, please indicate it, if this is not the case, specify the procedure used to determine the “ratios” (i.e. in the cases with >99 “ratio” of 3a how is this value determined?).

-          The authors justify the reaction time differences observed in Table 3 based on the “hindrance of the reaction center” (page 9, lines 268-272). Does the presence of substituents in para position increase the steric hindrance of the boronic acids for the Suzuki reaction?

-          Please provide the experimental procedure for preparation and characterization data for compounds 2, 5 and 6.

-          The purity of compounds 3b and 3e is not enough for their publication.

-          The baseline for the 31P NMR spectra of 3d is weird. Please provide a better spectra.

In addition the following minor points should be corrected:

Page 3, line 82 “CM-6” should be “CM-5”.

Page 4, line 140 change “-69%” to “69%”.

Scheme 1, indicate the catalyst load.

Page 5, lines 179-180, revise the sentence.

Scheme 4, change “eqv” to either “eq” or “equiv”. Homogenize throughout the manuscript.

Page 7, lines 217-219, revise the sentence. The meaning is not clear.

Page 8, please improve the graphical quality for the ORTEP representation in Figure 2.

Page 11, lines 312-315 and 322-326, the two sentence are repeated. Please erase one or the other.

Please add general structures showing the numbering used on the NMR description.

Page 21, line 620: the signal at 6.937 ppm should integer 2H.

In the characterization data for compounds 3d, 9h, 10d and 11d, 19F NMR data is missing.

In the 1H NNR and 13C NMR description for compound 11b there are 3 protons and 1 carbon missing, respectively.

In the supplementary material file the list of content is missing.

Author Response

We want to thank the Reviewer for the time spent on the detailed analysis of our work and comments on the manuscript.

Point 1: It is not clear how the authors have selected the substituents for the model compounds in the DFT study. A more systematic analysis would have been possible by a more judicious design. The authors should evaluate the CN in para and also the OMe in meta to have a better picture of the effects.

Answer: The revised version of the manuscript was rewritten according to the suggestions of the Reviewer. A section on the selection of the substituents was added as well as the additional DFT calculation data on the suggested models bearing -CN in para, -OMe in meta position and additionally a structure possessing -CN in meta position.

Point 2: The authors optimize the Suzuki reaction by testing different catalysts and solvents. Do the authors use an internal standard? If this is the case, please indicate it, if this is not the case, specify the procedure used to determine the “ratios” (i.e. in the cases with >99“ratio” of 3a how is this value determined?).

Answer: For the ratio of the products, we did not use determination with internal standard. The ratio of the compounds in the crude reaction mixture was determined using the integrals of the peaks for the H-4 in the coumarin species – 3a, 5, and 7. We based our research on that approach because we had a homogeneous sample, sufficient relaxation time for 1H NMR, good signal-to-noise ratio, and absence of other contributions to the chosen peaks. By applying this technique, only the ratio of the different species could be determined (relative concentration) and not the yields of the products (absolute concentration where an internal standard is needed).  The value of “>99” was listed for compound 3a when in the mixture only traces (small broad peaks viewed after enlargement of the spectra) of the starting material 5 was observed.

Point 3: The authors justify the reaction time differences observed in Table 3 based on the “hindrance of the reaction center” (page 9, lines 268-272). Does the presence of substituents in para position increase the steric hindrance of the boronic acids for the Suzuki reaction?

Answer: We agree with the reviewer that we made a mistake when we tried to summarize the results. The sentence was revised to “Thus, implies that the hindrance of the reaction center and the electronic effect of the groups play a major role in the outcome of the reactions.”

Point 4: Please provide the experimental procedure for preparation and characterization data for compounds 2, 5 and 6.

Answer: Compounds 2, 5 and 6 are well known compounds. The used experimental procedures were citated in Materials and Methods. A 1H NMR data was added.

Point 5: The purity of compounds 3b and 3e is not enough for their publication.

Answer: Additionally, a preparative HPLC-chromatography was performed on the products and better NMR spectra were added to the supporting information.

Point 6: The baseline for the 31P NMR spectra of 3d is weird. Please provide a better spectra.

Answer: A better NMR spectrum was added to the supporting information.

Point 7: In addition the following minor points should be corrected:

Page 3, line 82 “CM-6” should be “CM-5”.

Page 4, line 140 change “-69%” to “69%”.

Scheme 1, indicate the catalyst load.

Page 5, lines 179-180, revise the sentence.

Scheme 4, change “eqv” to either “eq” or “equiv”. Homogenize throughout the manuscript.

Page 7, lines 217-219, revise the sentence. The meaning is not clear.

Page 8, please improve the graphical quality for the ORTEP representation in Figure 2.

Page 11, lines 312-315 and 322-326, the two sentence are repeated. Please erase one or the other.

Page 21, line 620: the signal at 6.937 ppm should integer 2H.

In the 1H NNR and 13C NMR description for compound 11b there are 3 protons and 1 carbon missing, respectively.

Answer: We did a revision on the typographical and formatting errors.

Point 8: Please add general structures showing the numbering used on the NMR description.

Answer: The structures and their numbering were added to the NMR description section.

Point 9: In the characterization data for compounds 3d, 9h, 10d and 11d, 19F NMR data is missing.

Answer: Additional 19F NMR spectra were recorded for compounds 3d, 9d, 10d and 11d. Based on the desired spectra we assume that the reviewer meant compound 9d not 9h, because in the manuscript there is no compound listed as 9h

Point 10: In the supplementary material file the list of content is missing.

Answer: A list of content was added to the supplementary materials.